# *Imagine That!* Abstract-to-Intricate Text-to-Image Synthesis with Scene Graph Hallucination Diffusion

**Shengqiong Wu** [1]    **Hao Fei** [1*]  **Hanwang Zhang** [2]    **Tat-Seng Chua** [1]

[1]NExT++, School of Computing, National University of Singapore

[2] School of Computer Science and Engineering, Nanyang Technological University

swu@u.nus.edu   {haofei37, dcscts}@nus.edu.sg   hanwangzhang@ntu.edu.sg

## Abstract

In this work, we investigate the task of text-to-image (T2I) synthesis under the abstract-to-intricate setting, i.e., *generating intricate visual content from simple abstract text prompts*. Inspired by human imagination intuition, we propose a novel scene-graph hallucination (SGH) mechanism for effective abstract-to-intricate T2I synthesis. SGH carries out scene hallucination by expanding the initial scene graph (SG) of the input prompt with more feasible specific scene structures, in which the structured semantic representation of SG ensures high controllability of the intrinsic scene imagination. To approach the T2I synthesis, we deliberately build an SG-based hallucination diffusion system. First, we implement the SGH module based on the discrete diffusion technique, which evolves the SG structure by iteratively adding new scene elements. Then, we utilize another continuous-state diffusion model as the T2I synthesizer, where the overt image-generating process is navigated by the underlying semantic scene structure induced from the SGH module. On the benchmark COCO dataset, our system outperforms the existing best-performing T2I model by a significant margin, especially improving on the abstract-to-intricate T2I generation. Further in-depth analyses reveal how our methods advance.[2]

## 1   Introduction

The task of generating images from natural language descriptions, known as text-to-image (T2I) synthesis, has attracted significant attention [4, 17, 52]. To approach T2I, various generative models have been explored, including generative adversarial networks (GANs) [48, 52, 61], variational autoencoders (VAEs) [50], flow-based models [3], and auto-regressive models (ARMs) [10, 41], all of which aim to generate realistic images in high quality and high faithfulness. Most recently, diffusion-based models have been proposed, which simulate the physical process of gas diffusion for image generation [23]. Diffusion models have shown unprecedented performance in image synthesis over existing methods, becoming the current state-of-the-art (SoTA) T2I solution [2, 9, 19, 47, 42].

As a long-reached viewpoint [28, 53, 37], sound T2I systems should not only achieve high-quality image generation in simple straightforward visual scenery but be more capable of synthesizing realistic images with complex scenes. Typically, detailed textual descriptions are necessarily needed to prompt the synthesis process with adequate details for high-quality vision generation. However, in a realistic world, it could also be ubiquitous to produce intricate visions without relying on lengthy elaborate prompts. For example, users may prefer T2I systems to synthesize well-detailed images while not taking too much time to write descriptions in detail. More crucially, due to the natural modality asymmetry between language and vision, even some simple words can intrinsically describe or represent abstract visual scenes with rich and complex details. Whenever mentioning the words

---

[*]Hao Fei is the corresponding author.

[2]Code is available at https://github.com/ChocoWu/T2I-Salad

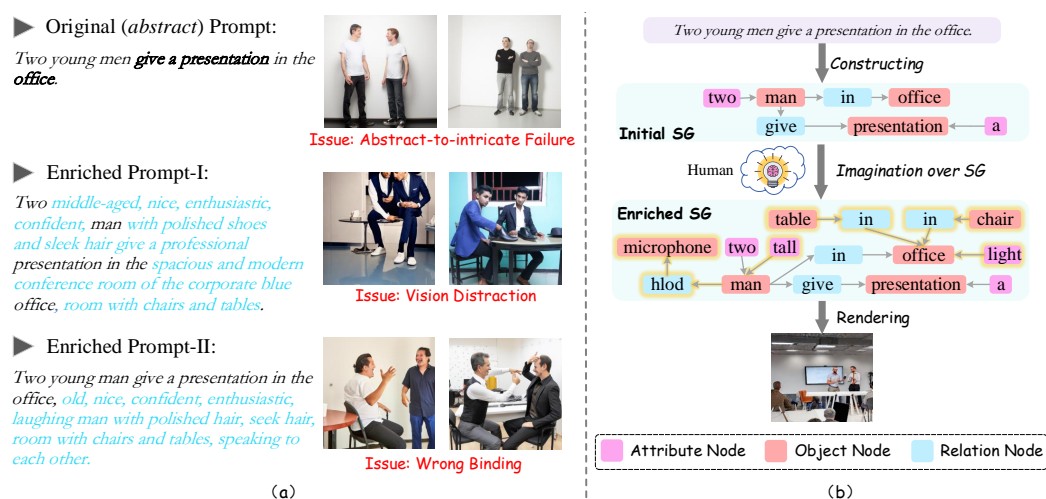

Figure 1: (a) An example of abstract-to-intricate T2I synthesis. All images are generated by the Latent diffusion Model (LDM) [42]. LDM fails to accurately render the abstract contexts, e.g., 'give a presentation' and 'office' of the original prompt. Raw prompts can be enriched via descriptive insertion [8], or addition [48]. Enriched contexts are in *blue*. (b) We illustrate the human intuition on the abstract-to-intricate T2I process: we always first grasp the semantic structure of the original prompt text, i.e., scene graph (SG), and then carry out imagination with more complete scenes based on the SG. Here the glowing nodes and edges are enriched ones.

with specific scenes, such as *classroom, kitchen, office*, or actional verbs e.g., *traveling, shopping*, there is always a picture with multifaceted scenes and rich-detailed backgrounds. In a word, *it is worth investigating generating intricate images from succinct abstract prompts*.

Yet existing prevailing approaches, even the SoTA diffusion models, may largely fail the abstract-to-intricate T2I, due to the lack of necessary details of input prompts (cf. Figure 1(a)). One intuitive workaround is to directly enrich the texts, i.e., adding more details for the prompts. Specifically, existing works either consider inserting additional adjectives and attributives to modify the original mentions and scenes [8], or concatenating raw sentences with more tangible explanations and contexts that are elicited from external large language models [48], e.g., ChatGPT [38]. Unfortunately, due to the intrinsic grammar and linguistics rules, such text-based prompt enrichment can be subject to the issue of lower controllability. One problem is the *visual distraction*, where the main focus of the resulting image is dominated by other newly-added trivial contents, when aggressively inserting intermediate descriptive components into the raw texts, as exemplified in Figure 1(a). Besides, directly appending new textual descriptions would increase the prompt length and then lead to *incorrect binding* of attributes or relations, i.e., making the image deviate from the original user intention.

As a reference, intuitively, we human beings would tackle the abstract-to-intricate T2I as a two-step painting process, i.e., from *semantic interpretation* to *scene imagination*. In the semantic interpretation, a painter always first translates the succinct textual prompt into a structured skeleton that represents the semantic scene of key mention objects and their relations. Then, based on the initial scene, the painter mentally completes the abstract scene with more concrete and valid details. With the enriched scene structure, the final vision can be more accurately and easily rendered. Motivated by such human intuition, in this work, we propose a **scene-graph hallucination** (SGH) method for achieving effective abstract-to-intricate T2I. As illustrated in Figure 1(b), we first investigate representing the input prompt with its scene graph (SG) [46]. SG advances in depicting the intrinsic semantics of texts (or vision) with structured representations, e.g., objects, attributes, and relationships, enabling fine-grained control of the semantic scene [53]. Based on the SG of input text we then carry out scene hallucination, expanding the initial SG with more possible specific scene structures. Also with the SG representation, the imagination process can be much more accurate and controllable.

To implement the overall idea, we develop an **S**G-based **h**allucination **d**iffusion system (namely, **Salad**) for high-quality image synthesis. Salad is a fully diffusion-based T2I system, which mainly consists of a scene-driven T2I module and an SGH module. As shown in Figure 2, we first take advantage of the SoTA latent diffusion model [9] as our backbone T2I synthesizer, in which the overt image-generating process is controlled and navigated by the underlying semantic scene structure.

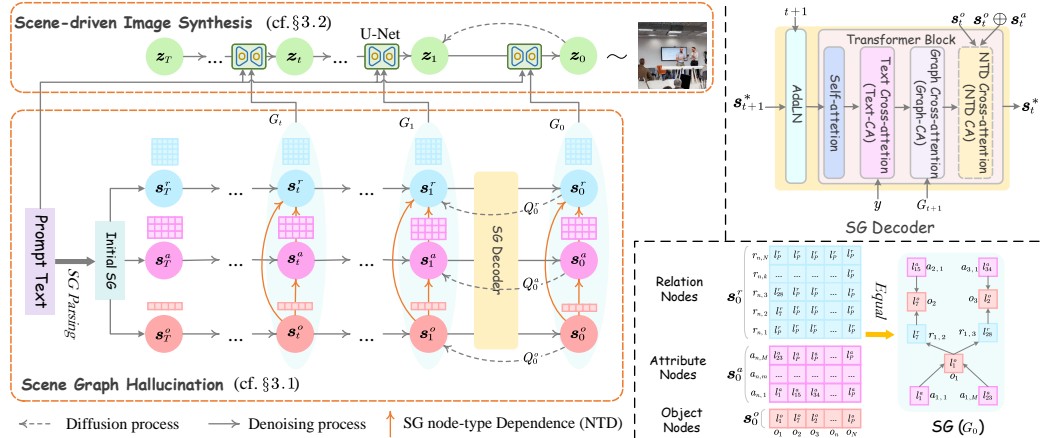

Figure 2: Overall framework of our proposed SG-based hallucination diffusion system (Salad).

On the other hand, we design the SGH module based on the modality-agnostic discrete diffusion model [25], which evolves and completes the initial SG structure of the input prompt by iteratively generating new scene elements (i.e., SG nodes). During the SG imagination process, the resulting structural representation at each step is fused into the T2I synthesizer via a hierarchical integration strategy. Further, we devise a scene sampling mechanism, via which the SGH module can generate various SG imaginations, and thus help achieve diversified image synthesis during inference.

We conduct experiments on COCO [33], the widely-used T2I dataset. Results show that Salad outperforms all the existing strong-performing T2I systems with significant margins. Further analysis reveals that modeling the SG structures helps synthesize high-quality images with stronger semantic controllability. Our proposed SGH mechanism is effective in inducing sound SG structures, helping produce more realistic images from short abstract text prompts. And the scene sampling strategy aids diversified T2I generation. In summary, this paper contributes in five aspects:

- We are the first to study the novel setup of intricate image synthesis from abstract texts.
- We solve the abstract-to-intricate T2I with a novel SG hallucination mechanism, which is implemented via discrete diffusion technique, performing scene enrichment with reasonable imagination.
- We propose a diffusion-based model with a hierarchical scene integration strategy for highly controllable and scalable image generation.
- We devise a scene sampling mechanism to generate various scene graphs for diversified image syntheses during inference.
- Our framework achieves new SoTA results in the abstract-to-intricate T2I generation.

## 2 Preliminary

### 2.1 Scene Graph Representation

The SG (denoted as $G$) [45] represents the semantic relationships among scene objects in a structure, where there are three types of nodes, i.e., **object**, **attribute**, and **relation**, cf. Figure 1(b). We formulate the object node set as $\{o_1, \cdots, o_N\}$, where $o_n$ denotes $n$-th object node; the attribute node set as $\{a_{1,1}, \cdots, a_{N,M}\}$, where $a_{n,m}$ means $m$-th attribute node of the $n$-th object node; the relation node set as $\{r_{1,1}, \cdots, r_{N,N}\}$, where $r_{i,j}$ means object node $o_i$ connects to the object $o_j$. All nodes come with a category label $l^{o/a/r}$, and each type of node has its own unique category vocabulary. For example as in Figure 2 right bottom, the object node $o_1$ with category label $l_1^o$ associated with two attribute nodes $a_{1,1}$ and $a_{1,M}$, with category label $l_1^a$ and $l_{23}^a$, respectively. And the object node $o_1$ connects to the object node $o_2$ through an edge $r_{1,2}$ with the category label $l_7^r$.

### 2.2 Diffusion Models

Diffusion models (DMs) [23] learn to convert a simple Gaussian distribution into a data distribution. Technically, DMs consist of a forward (diffusion) process and a reverse (denoising) process. In the forward process, the given data $x_0 \sim q(x_0)$ is gradually corrupted into an approximately standard normal distribution $x_T \sim p(x_T)$ over $T$ steps by increasingly adding noisy, formulated as

$q(\boldsymbol{x}_{1:T}|\boldsymbol{x}_0) = \prod_{t=1}^{T} q(\boldsymbol{x}_t|\boldsymbol{x}_{t-1})$. The learned reverse process $p_\theta(\boldsymbol{x}_{0:T}) = p(\boldsymbol{x}_T) \prod_{t=1}^{T} p_\theta(\boldsymbol{x}_{t-1}|\boldsymbol{x}_t)$ gradually reduces the noise towards the data distribution. To improve the fit of a generative model to the data distribution, a variational upper bound on the negative log-likelihood is optimized:

$$\mathcal{L}_{vlb} = \mathbb{E}_{q(\boldsymbol{x}_{1:T}|\boldsymbol{x}_0)} \left[ \log \frac{q(\boldsymbol{x}_T|\boldsymbol{x}_0)}{p_\theta(\boldsymbol{x}_T)} + \sum_{t=2}^{T} \log \frac{q(\boldsymbol{x}_{t-1}|\boldsymbol{x}_t, \boldsymbol{x}_0)}{p_\theta(\boldsymbol{x}_{t-1}|\boldsymbol{x}_t)} - \log p_\theta(\boldsymbol{x}_0|\boldsymbol{x}_1) \right], \tag{1}$$

where $p_\theta(\cdot)$ is estimated by a denoising network, which can be a time-conditional U-Net [43]. Recently, latent diffusion models (LDMs) [42, 14, 19] are introduced to adopt DMs to operate in an efficient, low-dimensional latent space, where a pre-trained encoder $\mathcal{E}$ maps the given data $\boldsymbol{x}_0$ into a latent code $z_0 = \mathcal{E}(\boldsymbol{x}_0)$, and a decoder reconstructs the final output image from the denoised latent $\mathcal{D}(\boldsymbol{z}_0) \sim \boldsymbol{x}_0$. Due to the higher computation sufficiency, this work thus takes the LDM as T2I backbone. Besides, DMs also have been extended to operate in discrete state spaces [1], i.e., performing diffusion and denoising processes over discrete variables, which have demonstrated competitive performances for discrete data, such as text [24] and layout [5]. Hence, we also adopt the discrete DMs to realize the SG induction process. Appendix §A.1 gives more technical details to the discrete diffusion models.

## 3 Methodology

Formally, T2I aims to generate an image $x$ that faithfully reflects the desired content in the input prompt text $y$. To approach abstract-to-intricate T2I, we propose an SG-based hallucination diffusion system (Salad), which is shown in Figure 2. The salad consists of two major modules. First, the SGH (cf. §3.1) is responsible for enriching the initial SG of the text prompt via a discrete diffusion model. Then, built upon an LDM (cf. §2.2), the Scene-driven Image Synthesis module (SIS, cf. §3.2) performs denoising for image synthesis, during which the derived SG features is fused via a hierarchical scene integration mechanism. The underlying SGH closely collaborates with the upper SIS at each step, and thus the semantic scene skeleton takes fine-grained control of the overt vision rendering. We also describe the optimization (cf. §3.3), and the scene sampling strategy (cf. §3.4).

### 3.1 Scene Graph Hallucination (SGH)

As aforementioned, we formulate the SGH as a discrete denoising diffusion process [25] (cf. Fig. 2). Specifically, in the forward process, the SG of the gold image, marked as $G_0$, will be corrupted into a sequence of increasingly noisy latent variables $G_{1:T} = \{G_1, G_2, \cdots, G_T\}$, where each SG node $s_{t,j}^* \in G_t, * \in \{o, a, r\}$ ($t$ is diffusion step, $j$ is the node index) takes a discrete value with $K^*$ category labels, and $o, a, r$ denotes the nodes' type, i.e., object ($o$), attribute ($a$), and relation ($r$). For simplicity, we omit subscripts $j$ and superscripts $*$ in the following description. The discrete diffusion process can be parameterized with a multinomial categorical transition matrix:

$$q(s_t|s_{t-1}) = \mathcal{B}^\top(s_t) \cdot \boldsymbol{Q}_t \cdot \mathcal{B}(s_{t-1}), \tag{2}$$

where $\mathcal{B}(s_t)$ denotes the column one-hot vector of $s_t$. And $\boldsymbol{Q}_t$ is the transition matrix, with $[\boldsymbol{Q}_t]_{ij} = q(s_t = j|s_{t-1} = i)$ representing the probabilities that $s_{t-1}$ transitions to $s_t$. Due to the property of the Markov chain, the cumulative probability of $s_t$ at arbitrary timestep from $s_0$ can be derived as $q(s_t|s_0) = \mathcal{B}^\top(s_t) \cdot \bar{\boldsymbol{Q}}_t \cdot \mathcal{B}(s_0)$, where $\bar{\boldsymbol{Q}}_t = \boldsymbol{Q}_t \boldsymbol{Q}_{t-1} \cdots \boldsymbol{Q}_1$. Inspired by [1, 19], we employ a *mask-and-replace* strategy to design the $\boldsymbol{Q}_t$. For each node $s_t$, we define three probabilities: 1) a probability of $\gamma_t$ to transition to [MASK] node, 2) a probability of $K\beta_t$ be resampled uniformly over all the $K$ categories, and 3) a probability of $\alpha_t = 1 - K\beta_t - \gamma_t$ to stay the same node. Notedly, the [MASK] node never transits to other states. Hence, the transition matrix $\boldsymbol{Q}_t$ can be formulated as[3]:

$$\boldsymbol{Q}_t = \begin{bmatrix} \alpha_t + \beta_t & \beta_t & \beta_t & \cdots & 0 \\ \beta_t & \alpha_t + \beta_t & \beta_t & \cdots & 0 \\ \vdots & \vdots & \vdots & \ddots & \vdots \\ \gamma_t & \gamma_t & \gamma_t & \cdots & 1 \end{bmatrix}. \tag{3}$$

The aforementioned discrete state-space models assume that all the standard nodes are switchable by corruption. However, as stated in §2.1, there are three different SG nodes under separate categories. Hence, we apply three disjoint corruption matrices $\boldsymbol{Q}_t^o \in \mathbb{R}^{K^o \times K^o}$, $\boldsymbol{Q}_t^a \in \mathbb{R}^{K^a \times K^a}$, $\boldsymbol{Q}_t^r \in \mathbb{R}^{K^r \times K^r}$ for object, attribute, and relation nodes, respectively, where $K^o, K^a, K^r$ denotes the size of category labels of three node types respectively.

---

[3]Appendix §A.1 provides detailed formulation of the discrete diffusion process.

In the denoising process, we introduce an SG decoder as the neural approximator to estimate the distribution $p_\theta(s_{t-1}|s_t, y)$. As shown in Figure 2, SG decoder first employs an adaptive normalization (AdaLN) to inject the timestep information. A text cross-attention (Text-CA) integrates the input prompt $y$. Then, a graph cross-attention (Graph-CA) is devised to take in the induced SG ($G_{t+1}$) in the previous $t+1$ timestep:

$$\tilde{\boldsymbol{H}}^* = \text{Graph-CA}(G_{t+1}, \boldsymbol{H}^*),\qquad(4)$$

where $\boldsymbol{H}^*$ are the features yielded from Text-CA. Graph-CA consistently consults the overall picture of the last SG for a more coherent generation when making the current decision.

Intuitively, among the object, attribute, and relation nodes, the object nodes always come first to determine the scene subjects, followed by their modifier attributes, and then the relations between objects. Thus, instead of parallel induction of three node types, we follow this SG node-type dependence (NTD) intuition, and design an NTD cross-attention (NTD-CA) for the $*$-type node induction ($*$ can be object or attribute):

$$\hat{\boldsymbol{H}}^* = \begin{cases} \text{NTD-CA}(s_t^o, \tilde{\boldsymbol{H}}^*), & * = a \\ \text{NTD-CA}(s_t^o \oplus s_t^a, \tilde{\boldsymbol{H}}^*). & * = r \end{cases}\qquad(5)$$

Note that we stack multiple layers of the above calculations as one SG decoder. For each state of node types $\hat{s}_t^*$, a softmax function is put on to obtain the category label distributions: $\hat{s}_t^* = \text{Softmax}(\hat{\boldsymbol{H}}^*)$.

Following [1], we optimize the SG decoder by minimizing the variational lower bound $\mathcal{L}_{vlb}$ (Eq. 1). Also the parameterization trick [19] is leveraged to encourage the system to predict the noiseless node distribution $p_\theta(\tilde{s}_0|s_t, y)$ at each reverse step, which is taken as an auxiliary learning objective to be incorporated with $\mathcal{L}_{vlb}$:

$$\begin{aligned} \mathcal{L}_{SGH} &= \mathcal{L}_{vlb} + \lambda_1\, logp_\theta(s_0|s_t, y), \\ \mathcal{L}_{vlb} &= \mathcal{L}_0 + \mathcal{L}_1 + \cdots + \mathcal{L}_{T-1} + \mathcal{L}_T, \\ \mathcal{L}_0 &= -logp_\theta(s_0|s_1, y), \\ \mathcal{L}_{t-1} &= D_{KL}((q(s_{t-1}|s_t, s_0))||(p_\theta(s_{t-1}|s_t, y))), \\ \mathcal{L}_T^o &= D_{KL}(q(s_T|s_0)||p_\theta(s_T)), \end{aligned}\qquad(6)$$

where $\lambda_1$ is a hyper-parameter for controlling the learning components.

## 3.2  Scene-driven Image Synthesis (SIS)

With the enriched SG from SGH in each denoising step at hand, the backbone T2I diffusion carries out the image synthesis with the guidance of that SG. We design a hierarchical scene integration (HSI) strategy to ensure the highly effective integration of SG features. Specifically, we consider the fusion at four different hierarchical levels, i.e., object ($L^o$), relation ($L^r$), region ($L^c$), and global levels ($L^g$) with each focusing on different context scopes, as illustrated in Figure 3. We maintain the representations of these three levels as the keys $\boldsymbol{K}_{L^i}$ & values $\boldsymbol{V}_{L^i}$ via CLIP encoder [40], which are then integrated together via the Transformer attention of U-Net in LDM:

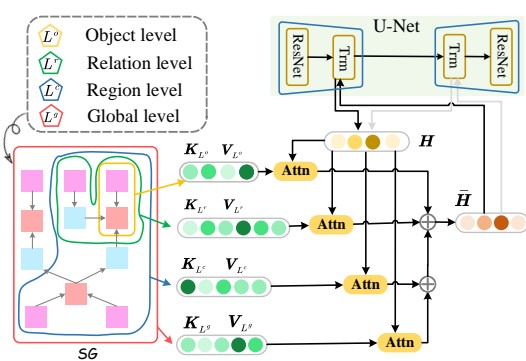

Figure 3: Hierarchical scene integration (HSI) fuses the SG features under multiple levels: 1) objects (with attributes), 2) relational triplets (i.e., *subject-predicate-object*), 3) regional neighbors, and 4) the whole SG.

$$\bar{\boldsymbol{H}} = \sum_{i\in\{o,r,c,g\}} \text{Attn}(\boldsymbol{H}, \boldsymbol{K}_{L^i}, \boldsymbol{V}_{L^i}) = \sum_{i\in\{o,r,c,g\}} \text{Softmax}(\frac{\boldsymbol{H}\boldsymbol{K}_{L^i}^\top}{\sqrt{d}})\boldsymbol{V}_{L^i},\qquad(7)$$

where $\boldsymbol{H}$ is the visual query vectors from the ResNet block in LDM. The above hierarchical scene integration is carried out for both the downsampling and upsampling processes in U-Net. By denoising $T$ steps, the system finally produces the desired image. Appendix §A.2 gives more details of this part.

## 3.3  Warm-start Training

To ensure stable learning of the overall system, we take a warm-start training strategy. Firstly, the SGH is separately updated via $\mathcal{L}_{SGH}$ (Eq. 6) based on the abstract-to-intricate SG pair annotations,

until it has converged. Then, both the SIS and SGH modules are optimized jointly by minimizing:

$$\mathcal{L} = \lambda_2 \mathcal{L}_{SGH} + \mathcal{L}_{SIS}, \quad \text{where } \mathcal{L}_{SIS} = \mathbb{E}_{\boldsymbol{z} \sim \mathcal{E}(\boldsymbol{x}_0), \epsilon \sim \mathcal{N}(0,\boldsymbol{I}), t}[||\epsilon - \epsilon_\theta(\boldsymbol{z}_t, G_t, y, t)||_2^2]. \tag{8}$$

Here we follow [23] to optimize SIS with a simple surrogate objective that calculates the mean-squared error loss, and $G_t$ is the intermediate SG by SGH at timestep $t$, which can be derived from the $s_t^*, * \in \{o, a, r\}$ (cf. Figure 2). $\epsilon$ is the noise in SIS, and $\epsilon_\theta(\cdot)$ denotes the U-Net (cf. Figure 3).

## 3.4 Inference with Scene Sampling

During inference, we further aim to endow the SGH with diversified SG enrichment, and thus lead to T2I diversification. Intuitively, given an abstract prompt, there is often more than one possibility of the potential scenes to imagine. Also, it can be observed that in the de-

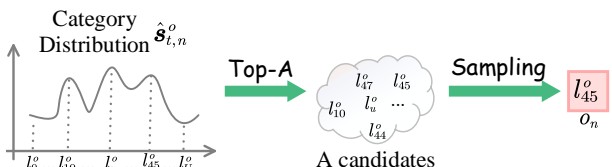

Figure 4: Illustration of the scene sampling mechanism.

noising process, the diffusion model has a larger potential of divergence only at its earlier stage, while the generation tends to be more stable and certain when the iteration grows. Correspondingly, we expect the scene sampling to start in the early denoising steps, and gradually be more determining. Thus, we design a scene-sampling mechanism. First, instead of picking the best prediction of the category of any node $\hat{s}_{t,j}^*$, we take the top-$A$ category candidates with corresponding probability distribution $\Psi$ based on the category distribution $\hat{s}_{t,j}^* \in \mathbb{R}^{K^*}$ of node $s_{t,j}^*$. Then, we perform sampling over these candidates with a dynamic probability:

$$\rho_{t,j}^* = e^{-\eta \cdot t} \cdot \Psi + (1 - e^{-\eta \cdot t}) \frac{1}{A}, \tag{9}$$

where $* \in \{o, a, r\}$, and $\eta$ is a temperature. It is an annealing process, i.e., when $t=T$ (starting denoising) more random sampling is preferable, while $t$ approaches 0 (denoising ends), SGH tends to be more decisive. Figure 4 exemplifies the mechanism with the *object* type of nodes ($\hat{s}_{t,n}^o$).

# 4 Experiments

## 4.1 Settings

**Data and Resource** We conduct T2I generation experiments mainly on the COCO [33] dataset. We also prepare the abstract-to-intricate SG pair annotations for training the SGH module, where we employ an external textual SG parser [46] and a visual SG parser [59] on the paired images and texts in COCO, to obtain the initial SG and imagined SG, respectively. To enlarge the abstract-to-intricate SG pairs, we further extend Visual Genome (VG) [30]. Besides, to simulate the abstract-to-intricate T2I scenario, we further manually extract a subset of text-image pairs from raw COCO data (named COCO-A2I), in which the texts are short and abstract,[4] while the images are comparatively complex and intricate. Appendix §B.1 shows all the dataset details.

**Baseline and Evaluation** We make comparisons with three types of existing strong-performing T2I models. 1) **GAN-based models**: AttnGAN [52], ObjGAN [32], DFGAM [48], OPGAN [22]. 2) **Auto-aggressive model**: DALL-E[41], and CogView [10]. 3) **Diffusion-based models**: LDM [42], VQ-diffusion [19], LDM-G and Frido [14] with classifier-free guidance. Note that LDM-G and Frido are the current SoTA T2I synthesizers. In addition, two types of **text-based enrichment approaches** are included as baselines: stable-diffusion prompt generator (SD-PG) and SPY inspired by [6]. The enriched text prompts are then utilized as inputs for Frido to generate the final images. We adopt three standard metrics to measure image synthesis performance: 1) **Inception score (IS)** [44], 2) **Fréchet Inception Distance (FID)** [21] and 3) **CLIP score**. Moreover, we use **GLIP** [31] to measure the fine-grained '*object-attribute*' grounding in images, and **Triplet Recall (TriRec.)** measure the '*subject-predicate-object*' triplet recall between two SGs. We also adopt the **Learned Perceptual Image Patch Similarity (LPIPS)** [60] for diversifying generation evaluation. Detailed definitions of evaluation metrics are shown in Appendix §B.2.

---

[4]We select the texts that mainly include two types of words, i.e., `Place Nouns` (e.g., office, airport, bathroom) and `Progressive Verbs` (e.g., traveling, knitting, gardening).

Table 1: The T2I results on the overall COCO dataset. ♭: taken from Fan et al. [14]; ♮: copied from Ding et al. [10], ♯: taken from Hinz et al. [22]. The best score is in bold, and the second best is underlined.

| Model | FID ↓ | IS ↑ | CLIP ↑ |
|---|---|---|---|
| ▶ **GAN Model** | | | |
| AttnGAN[♭] | 33.10 | 26.61 | - |
| ObjGAN[♭] | 36.52 | 24.09 | - |
| DFGAN[♭] | 21.42 | - | - |
| OPGAN[♯] | 24.70 | 27.88 | - |
| ▶ **Auto-aggressive Model** | | | |
| DALLE[♮] | 27.34 | 17.90 | - |
| CogView[♮] | 27.10 | 18.20 | - |
| ▶ **Diffusion Model** | | | |
| LDM[♭] | 17.61 | 19.34 | 65.00 |
| VQ-diffusion[♭] | 14.06 | 21.85 | 67.70 |
| LDM-G[♭] | 12.27 | 27.82 | 69.27 |
| Frido[♭] | 11.24 | 26.84 | 70.46 |
| **Salad** | **10.19** | **29.96** | **73.83** |

Table 2: Results on the COCO-A2I subset for the abstract-to-intricate T2I generation. 'SPY†' denotes enriched texts are parsed into SG, and then perform SG-to-image generation via Frido.

| Model | FID ↓ | IS ↑ | CLIP ↑ |
|---|---|---|---|
| ▶ **T2I Baseline** | | | |
| AttnGAN | 78.19 | 11.09 | 52.78 |
| ObjGAN | 75.33 | 13.16 | 55.20 |
| DFGAN | 71.24 | 15.56 | 56.91 |
| DALLE | 66.36 | 16.03 | 63.05 |
| CogView | 62.85 | 16.98 | 63.97 |
| LDM | 55.27 | 16.20 | 67.79 |
| VQdiffusion | 69.14 | 15.78 | 64.58 |
| Frido | 40.36 | 18.36 | 68.53 |
| ▶ **Text-based Enrichment (+Frido)** | | | |
| SD-PG | 36.50 | 17.64 | 65.23 |
| SPY | 35.59 | 21.59 | 67.86 |
| SPY† | 39.41 | 22.16 | 66.93 |
| **Salad** | **31.25** | **28.63** | **71.29** |

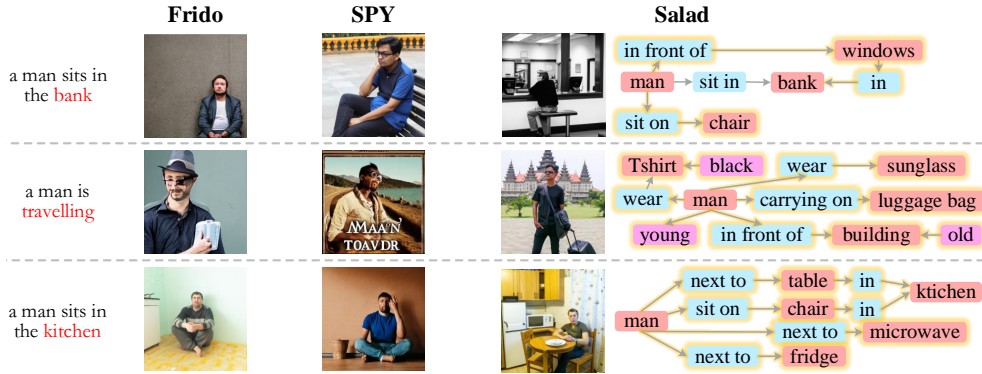

Figure 5: Qualitative results by different methods, where the given prompts randomly selected from the test set are short and abstract expressions (marked in red), while describing intricate visual scenes.

**Implementation** We define the maximum number of SG object nodes as 30, and each object node has a maximum of 3 attributes. We set the timesteps ($T$) for SGH and SIS as 100. For the SIS module, we load the parameters of Stable Diffusion[5] (v1.4) as the initialization. We use the CLIP[6] (vit-large-patch14) as our text encoder. We optimize the framework using AdamW [34] with $\beta_1 = 0.9$ and $\beta_2 = 0.98$. The learning rate is set to 5e-5 after 10,000 iterations of warmup. For the attention layer in SG decoder and UNet in SIS, we define a shared configuration as follows: 4 layers, 8 attention heads, 512 embedding dimensions, 2,048 hidden dimensions, and 0.1 dropout rate. We mainly follow the prior works [54, 18, 46] to acquire the visual scene graph (VSG) from the gold image and textual scene graph (TSG) from the text prompt.

### 4.2 Main Comparisons

Table 1 shows the main T2I generation results on the overall COCO data. We see that diffusion-based methods are consistently better than the other two types of T2I methods, especially on the FID metric. Most importantly, our proposed Salad model yields the overall best results on all metrics. For example, we outperform Frido by 1.05% FID, 3.37% on CLIP score, and surpass LDM-G by 2.1% IS score. This directly demonstrates the efficacy of our method.

To directly assess the capability of solving the abstract-to-intricate T2I synthesis, we further make comparisons on the collected COCO-A2I dataset. Also the two text-based enrichment methods are included. As shown in Table 2, with text enrichment, both SD-PG and SPY achieve better

[5]https://github.com/CompVis/stable-diffusion
[6]https://github.com/openai/CLIP

performance than the non-enriched T2I methods on both FID and IS metrics. However, we find that the two enrichment approaches fail to show superior results on the CLIP scores. This is mostly because text-based enrichment methods suffer the aforementioned issues of *wrong binding or visual distraction*, etc., which results in lower matching between the generated images and raw (abstract) texts. In contrast, our model presents the overall optimal performance, significantly. This directly verifies the effectiveness of our method with the scene graph hallucination mechanism, i.e., yielding reasonable imaginations of scenes and leading to high-quality abstract-to-intricate T2I synthesis.

### 4.3 Ablation Studies

Here we present the model ablations to quantify the contribution of each part of our system. The results are shown in Table 3. First, canceling the whole SGH module (w/o both SG guidance and imagination), the most significant performance drops are witnessed, indicating the pivot influence of the SGH mechanism. Also, the drops from 'SGH→Init.SG' directly reveal the importance of our scene enrichment mechanism. Interestingly, by comparing the ablating drop in COCO and COCO-A2I, we can notice that the SGH mechanism is especially more important under the abstract-to-intricate setup, while the SG features more stand out for the general T2I scenario. Besides, both properly modeling SG features during denoising and modeling the SG node-type dependence (NTD) contribute to the overall

Table 3: Ablation results (FID↓). 'SGH→Init.SG': using the initial SG of text without SG enrichment throughout the whole T2I generation process. 'w/o SGH': removing the overall SGH module, i.e., also meaning both without scene enrichment and SG guidance. 'HSI→GCN': encoding the SG with a GCN [35] instead of our HSI mechanism.

| Item | COCO | COCO-A2I |
|---|---|---|
| Salad (Ours) | **10.19** | **31.25** |
| SGH→Init.SG | 12.45(+2.26) | 37.77(+6.52) |
| w/o SGH | 16.98(+6.79) | 34.32(+3.07) |
| w/o Graph-CA (Eq.4) | 10.95(+0.76) | 32.74(+1.49) |
| w/o NTD-CA (Eq.5) | 11.19(+1.00) | 34.35(+3.10) |
| HSI→GCN | 10.87(+0.68) | 35.96(+1.71) |

performance. Further, when we abandon the hierarchical manner of the scene feature integration (HSI) and instead use a GCN to encode the overall SG, there are also clear performance drops.

### 4.4 Qualitative Results

To gain a more intuitive understanding of our model's capability on the abstract-to-intricate T2I synthesis, here we show some qualitative results. We visualize the generated images by different methods, where the given input prompts are in short and abstract format but describe intricate visual scenes. As shown in Figure 5, we find that Frido fails to generate images that precisely reflect the prompt instructions. In contrast, the text-based enriched method, SPY, yields visual results with much more details, in which the visual semantics, unfortunately, deviates much from the raw inputs largely. Overall, Salad is able to produce high-quality images with rich visual scenes and sophisticated contexts, meanwhile ensuring semantic accuracy, i.e., coinciding with the abstract input texts.

### 4.5 Analyses and Discussions

Via the above experiments, we have demonstrated the technical efficacy of our model. Following we further explore how our methods advance.

▶ **Q1: *How does SG guidance aid the generation of high-quality images?*** We first consider assessing the overall semantic matchness between the input texts and the generated images. We make comparisons with baselines without integrating SGs. By observing the CLIP scores in Figure 6, it is clear that our model can achieve higher semantics faithfulness of image synthesis. Further, we try to probe the semantic structural alignment between the inputs and outputs. We mainly measure the '*object-attribute*' correspondence between input texts and generated images via GLIP metric; and also assess if the structural triplets in initial textual SG can also be retrieved in the SG of the generated image (i.e., TriRec. metric). As shown in the

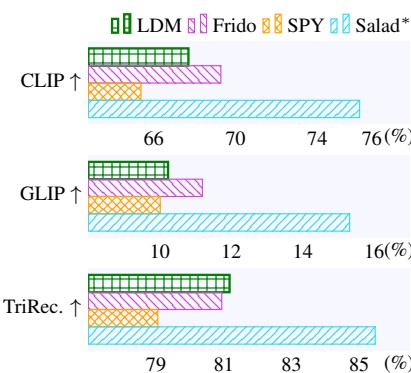

Figure 6: Evaluating the matches between input texts and generated images on COCO. For comparison, our 'Salad*' is downgraded by removing the SG imagination ability (i.e., SGH→Init.SG).

figure, our Salad system consistently performs much better than the baselines in terms of these two metrics. This significantly suggests that with the SG guidance, the structural controllability of the T2I process is greatly enhanced, and thus leads to high-quality generation.

► **Q2:** *Does SGH indeed induce intricate and reasonable SGs?* Firstly, in Figure 7 we explore whether the SGH can produce new SG structures during the T2I process, on the COCO and COCO-A2I datasets separately. As seen, after scene enrichment, the average numbers of all three element types (object, attribute, and relation) substantially increase. Notably, the addition is more evident on the COCO-A2I set, where scene enrichment is more needed.

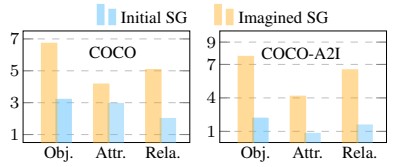

Figure 7: The average number of three types of SG nodes.

Next, we examine if the newly imagined SG structures provide reasonable scenes to the input prompt. We reach this by measuring the recall rate (TriRec.) of the '*subject-predicate-object*' pairs between two SGs, i.e., the predicted and the gold SGs. Assuming the SGs (denoted as $G^I$) of gold images entail reasonable scenes, we consider two types of SGs from our 'prediction': 1) SGs induced by SGH ($\overline{G}$), and 2) SGs parsed from our synthesized images ($G^{\overline{I}}$). We also compare with the text-enriched SPY method. As shown in Table 4, we first observe that the induced SGs highly align with the gold SGs, with 82.01 TriRec. score, indicating the

Table 4: Comparing the glod SGs ($G^I$) with our induced SG ($\overline{G}$), and the SG ($G^{\overline{I}}$) of generated image with TriRec. metric.

|  | $\overline{G}$ *vs.* $G^I$ | $G^{\overline{I}}$ *vs.* $G^I$ |
|---|---|---|
| SPY | - | 78.61 |
| Salad | 82.01 | 86.04 |

induced SGs are sensible for generating high-quality images. Moreover, by comparing the TriRec. scores ($G^{\overline{I}}$ *vs.* $G^I$) between SPY and Salad (i.e., 78.61 vs. 86.04), we learn that the synthesized images coordinated by imagined SGs more correspond to the gold images in terms of semantic scene structure. This also suggests that the SGH can induce valid SGs which favorably guide the image generation process.

► **Q3: Does sampling strategy helps diversified T2I?**
To measure the effectiveness of the sampling strategy in diversifying image generation given the same prompt, we consider both the qualitative analyses and human evaluation, where the former calculates the LPIPS score [60] to assess the perceptual similarity between two images in deep feature space. As shown in Figure 8, our Salad model significantly outperforms LDM and Frido in terms of diversity score and human evaluation, and things go worse upon removing the scene sampling

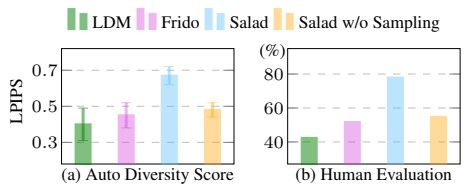

Figure 8: Comparison of diversity using diversity score (LPIPS) and human evaluation.

strategy. This demonstrates that our proposed scene sampling mechanism is effective in helping diversified T2I. In Appendix §C.2 we show more examples of diversified generations via the scene sampling mechanism.

## 5  Related Work

T2I is a long-standing topic in computer vision and multimodal communities. Arrays of explorations have been devoted to achieving stronger image synthesis performances with various deep generative models, such as GANs [4, 17, 48, 52, 61], VAEs [29], flow-based approaches [11, 12] and ARMs [7, 13, 39, 49]. More recently, the diffusion denoising probabilistic methods (DDPMs) have revealed the greatest potentials on image synthesis, in which the optimal density estimation is more naturally achieved with a fixed diffusion process to transform an image into a Gaussian noise [9, 19, 23, 42, 47]. This work follows the line of diffusion methods and takes the SoTA latent diffusion model (LDM) [42, 14, 19] as the T2I backbone.

Generating high-quality images with complex scenes is the key criterion of a sound T2I system [14, 15, 26]. Many efforts have been paid for synthesizing more realistic and nature-looking images in sophisticated scenes, yet most of which are conditioned on taking the detailed descriptions as inputs [6, 51]. Thus, how to generate high-quality images in intricate scenes from succinct and abstract prompts becomes a meaningful yet challenging task. In this work, we introduce a scene hallucination mechanism, which, built upon the SG structure, performs more accurate and controllable scene completion and eventually helps generate intricate images of higher quality. We consider the SG representations [27] for the input prompt texts as well as the guidance of image synthesis. SG advances in intrinsically describing the semantic structures of scenes for texts or images [28, 53], enabling more fine-grained control of complex scenes [53], and thus aiding the final image generation.

Scene graph (SG) is a type of structured data that represents multiple objects and their complex relationships in the vision or language scenes, wherein the nodes denote objects & attributes and the edges depict relationships between objects [27]. As intrinsically describing the semantic structures of scene semantics for the given texts or images, SG has been widely utilized as a type of external feature being integrated into downstream applications for enhancements, e.g., image retrieval [27, 56], image generation [28, 53] and image captioning [54, 36, 55]. In this work, we consider the SG representations for the input sentences. Compared to the linear sequential nature of the text, SG offers a more intuitive manner to represent the scene semantics in a structured format, enabling more fine-grained control of complex scenes [53], and thus aiding the final image synthesis.

Our SGH mechanism also relates to the research of SG enrichment or imagination. While existing methods mostly approach the task by incrementally parsing SG elements [16, 57, 58], such greedy-increment paradigm may largely suffer from trapping in locally optimal SG generation, thus leading to inferior SG imagination. Instead, in this paper, we implement SGH as a discrete denoising&diffusion process. Discrete diffusion technique [25] is the latest introduced method that replaces the continuous state in standard diffusion models with a discrete one. During each denoising step, the entire SG structure is updated and optimized from a global viewpoint, so as to yield a more reasonable enrichment of scene structure. Besides, both the T2I and the SGH are modeled as the same diffusion process in our framework, where the two processes are well synchronized, such that the underlying SG features can perfectly guide the T2I synthesis at each step. To our knowledge, we are the first to investigate SG induction using discrete diffusion models.

## 6 Conclusion

In this work, we explore the text-to-image synthesis task under the abstract-to-intricate setup. Drawing inspiration from human intuition, we propose a scene-graph hallucination mechanism, which carries out scene imagination based on the initial scene graph of the input prompt, expanding the starting SG with more possible specific scene structures. We then develop an SG-based hallucination diffusion system for the abstract-to-intricate T2I, which mainly includes an SG-guided T2I module and an SGH module. Specifically, we design the SGH module based on the discrete diffusion technique, which evolves the initial SG structure by iteratively adding new scene elements. Then, we utilize another continuous diffusion model as the T2I synthesizer, where the overt image-generating process is navigated by the underlying semantic scene structure induced by the SGH module. On the standard COCO dataset, our system shows great superiority in the abstract-to-intricate T2I generation. Further analyses demonstrate that our SG-based hallucination mechanism is able to generate logically sound SG structures, which in return helps produce high-quality scene-riched images.

## 7 Broader Impact

**Benefits** The current text-conditioned image generation approaches largely fail to the abstract-to-intricate T2I due to a lack of necessary details of input prompts. In this work, we propose a novel scene-graph hallucination mechanism inspired by human imagination intuition, which expands upon the initial scene graph from the text prompts to generate more feasible and specific scene structures. Furthermore, the enriched timestep-wised SG is leveraged to navigate the T2I generation process, leading to synthesizing more intricate images. Our study demonstrates that hallucinating images based on scene graph structures offer scalability, and modeling these structures enhances the generation of high-quality images with improved semantic controllability.

**Potential weakness** There can be two potential weaknesses that warrant consideration in our system. Firstly, the effectiveness of our system relies heavily on the quality of scene graph hallucination (SGH), yet the absence of a dedicated dataset for the SGH task poses a challenge in training the SGH module. However, we can leverage the richly annotated Visual Genome (VG) dataset, commonly used for training visual SG parsers, to provide initial training for the SGH module under an unconditional setting. Secondly, the training process of a diffusion model for text-to-image (T2I) generation entails substantial computational resources, resulting in increased energy consumption, $CO_2$ emissions, and potential environmental pollution.

## Acknowledgements

This research is supported by NExT++ Lab and CCF-Baidu Open Fund.

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

# A  Extension of Technical Details

We in this part extend the specific details of our method techniques.

## A.1  Discrete Diffusion Model for Scene Graph Hallucination

Here, we detail the forward and reverse processes in the discrete diffusion model [1] for SGH.

**Forward Process**   In the forward process, we consider a node $s_{t,j}^* \in G_t, * \in \{o, a, r\}$ ($t$ is diffusion step, $j$ is the node index) takes a scalar discrete value with $K^*$ categories, $s_{t,j}^* \in 1, \cdots, K^*$. Without introducing confusion, we omit subscripts $j$ and superscripts $*$ in the following description. We define the probabilities that $s_{t-1}$ transits to $s_t$ using the matrices $[\boldsymbol{Q}_t]_{ij} = q(s_t = j | s_{t-1} = i)$, then we can write the forward Markov diffusion process:

$$q(s_t | s_{t-1}) = \mathcal{B}^\top(s_t) \boldsymbol{Q}_t \mathcal{B}(s_{t-1}), \tag{10}$$

where $\mathcal{B}(s_t)$ denotes the one-hot column vector which length is $K$. The categorical distribution over $s_t$ is given by the vector $\boldsymbol{Q}_t \mathcal{B}(s_{t-1})$. Due to the property of the Markov chain, starting from $s_0$, we can derive the cumulative probability of $s_t$ at arbitrary $t$-timestep:

$$q_{(s_t | s_0)} = \mathcal{B}^\top(s_t) \cdot \bar{\boldsymbol{Q}}_t \cdot \mathcal{B}(s_0), \text{ where } \bar{\boldsymbol{Q}}_t = \boldsymbol{Q}_t \boldsymbol{Q}_{t-1} \cdots \boldsymbol{Q}_1. \tag{11}$$

Besides, by conditioning on $s_0$, the posterior of this diffusion process is tractable:

$$q(s_{t-1} | s_t, s_0) = \frac{q(s_t | s_{t-1}, s_0) q(s_{t-1} | s_0)}{q(s_t | s_0)} = \frac{(\mathcal{B}^\top(s_t) \boldsymbol{Q}_t \mathcal{B}(s_{t-1}))(\mathcal{B}^\top(s_{t-1}) \bar{\boldsymbol{Q}}_t \boldsymbol{v}(s_0))}{\mathcal{B}^\top(s_t) \bar{\boldsymbol{Q}}_t \mathcal{B}(s_0)}. \tag{12}$$

Note that the transition matrix $\boldsymbol{Q}_t$ is capable of controlling the data corruption and denoising process, thus it should be carefully designed such that it is not too difficult for the reverse network to recover the signal from noises. We follow [1, 19] that exploits a *mask-and-replace* strategy to design a $\boldsymbol{Q}_t$, which can be defined as:

$$\boldsymbol{Q}_t = \begin{bmatrix} \alpha_t + \beta_t & \beta_t & \beta_t & \cdots & 0 \\ \beta_t & \alpha_t + \beta_t & \beta_t & \cdots & 0 \\ \vdots & \vdots & \vdots & \ddots & \vdots \\ \gamma_t & \gamma_t & \gamma_t & \cdots & 1 \end{bmatrix}, \tag{13}$$

where for each node $s_t$, there are three probabilities, i.e., a probability of $\gamma_t$ to transition to [MASK] node, a probability of $K\beta_t$ be resampled uniformly over all the $K$ categories, and a probability of $\alpha_t = 1 - K\beta_t - \gamma_t$ to stay the same node. Notedly, the [MASK] node never transitions to other states. The aforementioned discrete state-space models assume that all the nodes are switchable by corruption. However, it is unreasonable that an object node, 'bed', transitions to a relation node, 'in'. To avoid such invalid transition, we propose to apply disjoint corruption matrices $\boldsymbol{Q}_t^o, \boldsymbol{Q}_t^a, \boldsymbol{Q}_t^r$ for object, attribute, and relation nodes, respectively.

**Reverse Process**   In the reverse process, an SG decoder is introduced as a denoising network to estimate the posterior distribution $p_\theta(\cdot)$, which takes the node token $s_t$, time step $t$ and text prompt $y$. Specifically, each layer of the SG decoder contains two parts: 1) an adaptive normalization (AdaLN), and 2) a transformer block. AdaLN is applied to inject the timestep information:

$$\boldsymbol{H}^* = \text{AdaLN}(s_t, t) = \boldsymbol{a}_t \text{LayerNorm}(\text{CLIP}(s_t)) + \boldsymbol{b}_t, \tag{14}$$

where $\text{CLIP}(\cdot)$ denotes an encoding layer for projecting the node token, and $\boldsymbol{a}_t$ and $\boldsymbol{b}_t$ are obtained from a linear projection of the timestep embedding. Each transformer block contains 1) a full self-attention, i.e., $\boldsymbol{H}^* = \text{Softmax}(\frac{\boldsymbol{H}^* \cdot \boldsymbol{H}^{*\top}}{\sqrt{d_1}}) \cdot \boldsymbol{H}^*$. 2) a text-cross-attention (Text-CA) integrates the input prompt $y$:

$$\boldsymbol{H}^* = \text{Text-CA}(y, \boldsymbol{H}^*) = \text{Softmax}(\frac{\boldsymbol{H}^* \cdot \boldsymbol{H}^{y\top}}{\sqrt{d_1}}) \cdot \boldsymbol{H}^y, \tag{15}$$

where $\boldsymbol{H}^y$ is a conditional feature sequence yielded by a text encoder first takes the text prompts $y$ as input. Then, 3) a graph cross-attention (Graph-CA) is devised to take the induced SG ($G_{t+1}$) in the previous $t + 1$ timestep:

$$\tilde{\boldsymbol{H}}^* = \text{Graph-CA}(G_{t+1}, \boldsymbol{H}^*) = \text{Softmax}(\frac{\boldsymbol{H}^* \cdot \boldsymbol{H}^{G\top}}{\sqrt{d_1}} \cdot \boldsymbol{E}^G) \cdot \boldsymbol{H}^G, \tag{16}$$

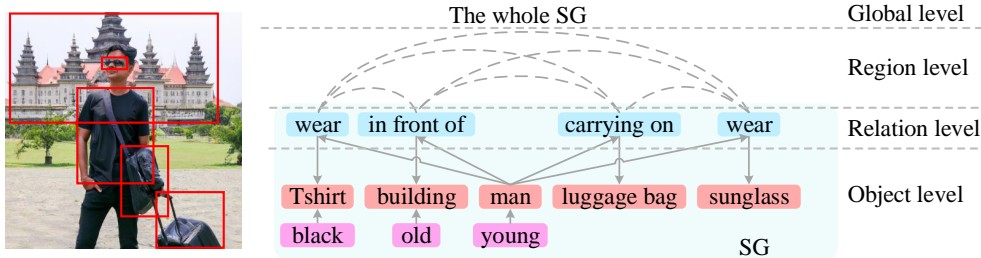

Figure 9: Different semantic levels in the SG: objects, relation, region, and global levels.

where $\boldsymbol{H}^G$ and $\boldsymbol{E}^G$ is the embedding representation of the nodes and edges in $G_{t+1}$, respectively, Next, 4) a node-type dependency cross-attention (NTD-CA) is designed for the $*$-type node induction (* can be object or attribute):

$$
\hat{\boldsymbol{H}}^* = \begin{cases} \text{NTD-CA}(s_t^o, \tilde{\boldsymbol{H}}^*) = \text{Softmax}(\frac{\tilde{\boldsymbol{H}}^* \cdot (\text{CLIP}(s_t^o))^\top}{\sqrt{d_1}}) \cdot \text{CLIP}(s_t^o), & * = a \\ \text{NTD-CA}(s_t^o \oplus s_t^a, \tilde{\boldsymbol{H}}^*) = \text{Softmax}(\frac{\tilde{\boldsymbol{H}}^* \cdot (\text{CLIP}(s_t^o) \oplus \text{CLIP}(s_t^a))^\top}{\sqrt{d_1}}) \cdot (\text{CLIP}(s_t^o) \oplus \text{CLIP}(s_t^a)). & * = r \end{cases}
$$
(17)

Finally, a softmax function is put on to obtain the category label distributions:

$$\hat{s}_t^* = \text{Softmax}(\hat{\boldsymbol{H}}^*).$$
(18)

Following [1, 19], the SG decoder is optimized by minimizing the variational lower bound (VLB):

$$\mathcal{L}_{vlb} = \mathcal{L}_0 + \mathcal{L}_1 + \cdots + \mathcal{L}_{T-1} + \mathcal{L}_T,$$
(19)

$$\mathcal{L}_0 = -log p_\theta(s_0|s_1, y),$$
(20)

$$\mathcal{L}_{t-1} = D_{KL}((q(s_{t-1}|s_t, s_0))||(p_\theta(s_{t-1}|s_t, y)),$$
(21)

$$\mathcal{L}_T^o = D_{KL}(q(s_T|s_0)||p_\theta(s_T)),$$
(22)

where $p_\theta(s_T)$ is the stationary distribution at timestep $T$. To estimate the $p_\theta(s_T)$, we extend the forward process by appending a rank-one matrix $\boldsymbol{Q}_{t+1}$ that ignores $s_T$ and produces a deterministic $s_{T+1} \in G_{T+1}$ where $G_{T+1}$ is the initial SG parsed from the text prompt $y$, then learning the reverse step $p_\theta(s_T|s_{T+1}) = p_\theta(s_T)$. Furthermore, we follow the reparameterization trick proposed in [19], which lets the network predict the noiseless token distribution $p_\theta(\tilde{s}_0|s_t, y)$ at each reverse step so as to gain better quality. Based on the reparameterization trick, an auxiliary objective is introduced:

$$\mathcal{L}_{s_0} = -log p_\theta(s_0|s_t, y),$$
(23)

To combine this loss with $\mathcal{L}_{vlb}$, the final training objects are defined as follows:

$$\mathcal{L}_{SGH} = \mathcal{L}_{vlb} + \lambda_1 \mathcal{L}_{s_0},$$
(24)

where $\lambda_1$ is a hyper-parameter to control the effect of the auxiliary loss $\mathcal{L}_{s_0}$.

## A.2 Scene-driven Image Synthesis

As described in §3.2, we propose a hierarchical scene integration strategy to effectively integrate the SG features. Here, we give more details on this part. Firstly, as shown in Figure 9, the SG is able to represent four different hierarchical levels of semantics corresponding to the image:

- **Object level**. In the process of image synthesis, a crucial aspect lies in accurately generating each specified individual object, corresponding to the object node and associated attribute notes in SG, for example, *black Tshirt*, *old building*, *young man*, *luggage bag*, *sunglass*.
- **Relation level**. A high-quality image is not only the generation of objects but also their intricate relationships which connect two objects, akin to the *subject-predicate-object* triplets found in the SG, such as *man wear Tshirt*, *man in front of building*, *man carrying on luggage bad*, *man wear sunglass*.
- **Region level**. Regional image generation focuses on multiple objects and entangled relationships among them, which aligns with the presence of overlapping relation triplets[7] in SG. Here we adopt two overlapped relation triplets as the region representation of the image, such as *man wear Tshirt in front of building*, *man wear Tshirt carrying on luggage bag*, *man wear*

---

[7]Two overlapped relation triplets means there are the same subject or object in the two relation triplets.

*Thirt wear sungalss*, *man in front of building carrying on luggage bag*, *man in front of building wear sunglass*, *man carrying on luggage bag wear sunglass*.

- **Global level** . The whole SG provides global semantics to guide the generation of images.

Then, we technically extract a collection of concepts from the four semantics levels in the SG, denoted as $L^o$ (object level), $L^r$ (relation level), $L^c$ (region level), and $L^g$ (global level). We encode each concept separately:

$$\boldsymbol{U}_{L^i} = \{\boldsymbol{u}_1, \boldsymbol{u}_2, \cdots\}, \ \boldsymbol{u}_j = \text{CLIP}(c_j),$$
$$\text{where } c_j \in L^i \, ; j = 1, 2, \cdots, |L^i| \, ; i \in \{o, r, c, g\} \tag{25}$$

We further maintain the representation of these four levels as the keys $\boldsymbol{K}_{L^i}$ and values $\boldsymbol{V}_{L^i}$ by linear transformations:

$$\boldsymbol{K}_{L^i} = \text{Linear}(\boldsymbol{U}_{L^i}); \ \boldsymbol{V}_{L^i} = \text{Linear}(\boldsymbol{U}_{L^i}), \tag{26}$$

where $i \in \{o, r, c, g\}$. Next, we integrate these features via the Transformer attention of U-Net in LDM, which is formulated in Eq. (7).

# B  Detailed Experiment Settings

## B.1  Datasets

**COCO**   The training and validation data numbers in COCO are 83K and 41k, respectively. We note that, in the evaluation phase, models are evaluated on the full COCO 2014 validation set.

**Visual Genome (VG)**   Visual Genome [30] version 1.4 (VG) comprises 108,077 images annotated with scene graphs. Following previous work [28], we use object and relationship categories occurring at least 2,000 and 500 times respectively in the dataset, leaving 178 objects and 45 relationship types, and we ignore small objects, and use images with between 5 and 30 objects and at least three relationships, this leaves us with 62,565 images with an average of 10 objects and 5 relationships per image.

**Construction of COCO-A2I**   We elaborate on the process of constructing the **COCO-A2I** dataset in the following three steps:

- First, we consider `Place Nouns` and `Progressive Verbs` are two types of abstract words that can depict intricate images. Therefore, we pre-define the candidate list of Place Nouns: *street*, *sidewalk*, *kitchen*, *restroom*, *bathroom*, *living room*, *bedroom*, *hostel*, *house*, *office*, *bank*. and the candidate list of Progressive verbs: *traveling*, *knitting*, *gardening*, *shopping*, *presenting*, *drawing*, *baking*, *studying*, etc.
- Second, we select the COCO valid dataset in that captions of instances contain the words in the candidate list, obtaining the primary-filtering dataset.
- Third, we filter out the primary-filtering dataset in which the number of words in the captions of instances is more than 10 and the number of bounding boxes is less than 6, obtaining the final target COCO-A2I dataset.

After the three-step pipeline, we obtain 2,005 text-image pairs.

In Figure 10, we show some examples, where the images tend to contain intricate content, including multiple objects, attributes, and relationships while the corresponding text prompts are relatively short and abstract.

## B.2  Evaluation Metric Implication

We employ **Fréchet Inception Distance (FID)** [21], **Inception score (IS)** [44], **CLIP score** [20], and **GLIP** [31] used in [14] to quantitatively evaluate the quality of the generated images, and **Learned Perceptual Image Patch Similarity (LPIPS)** [60] utilized in [42] to quantify the diversity of the generated images. Additionally, we introduce **Triplet Recall (TriRec.)** to measure the percentage of the correct relation triplet among all the relations. Technically, given a set of ground truth triplets (*subject-predicate-object*), denoted $GT$, and the TriRec. is computed as TriRec.$= |PT \cap GT|/|GT|$, where $PT$ are the relation triplets extracted from the generated images by a visual SG parser.

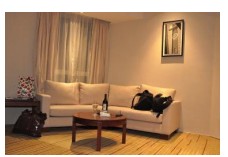
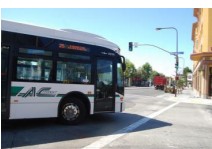
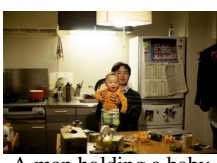
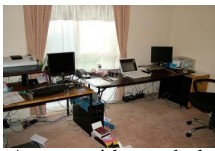

A few bags laying around in a living room.

A bus turning a corner on a city street.

A man holding a baby up in a kitchen.

A room with two desk covered in computer equipment.

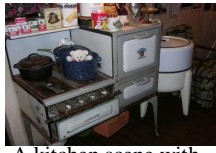
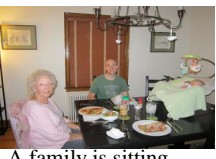
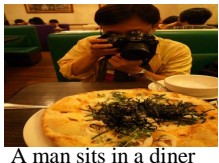
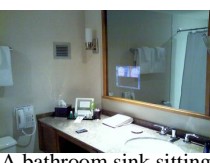

A kitchen scene with an oven and a stove.

A family is sitting around a dinner table.

A man sits in a diner photographing his meal.

A bathroom sink sitting underneath a mirror in a bathroom.

Figure 10: The examples from COCO-A2I dataset for evaluating abstract-to-intricate T2I generation.

## B.3 Human Evaluation Criterion.

In §4.5, we conduct a human evaluation, i.e., user study, for more intuitive assessments of the diversification quality of T2I models. Specifically, we generate 10 images conditioned on a single text prompt and ask evaluators to cluster these images based on their contextual information, such as objects, attributes, and relationships present in the generated images. After evaluating 100 distinct text prompts, we compute the average clustering ratio:

$$\textit{Average Clustering Ratio} = \text{Average}(\frac{|Cluster|}{10}, \cdots), \qquad (27)$$

where $|Cluster|$ denotes the number of clusters for 10 images generated by a text prompt. Average($\cdot$) is an average function. Here, a higher ratio score signifies greater diversification in image generation.

## C   More Experiment Results

### C.1   Visualizations of SGH Process

In Figure 11, we visualize the SGH process with two examples, where we sample several time steps and plot the generated SGs and the corresponding images. As can be seen, our SGH is capable of imaging certain reasonable structures of SGs to enrich the initial SG.

### C.2   More Cases

Here we showcase more examples of the generated images via the scene sampling mechanism in Figure 12 and 13. We synthesize 10 images for each given text prompt.

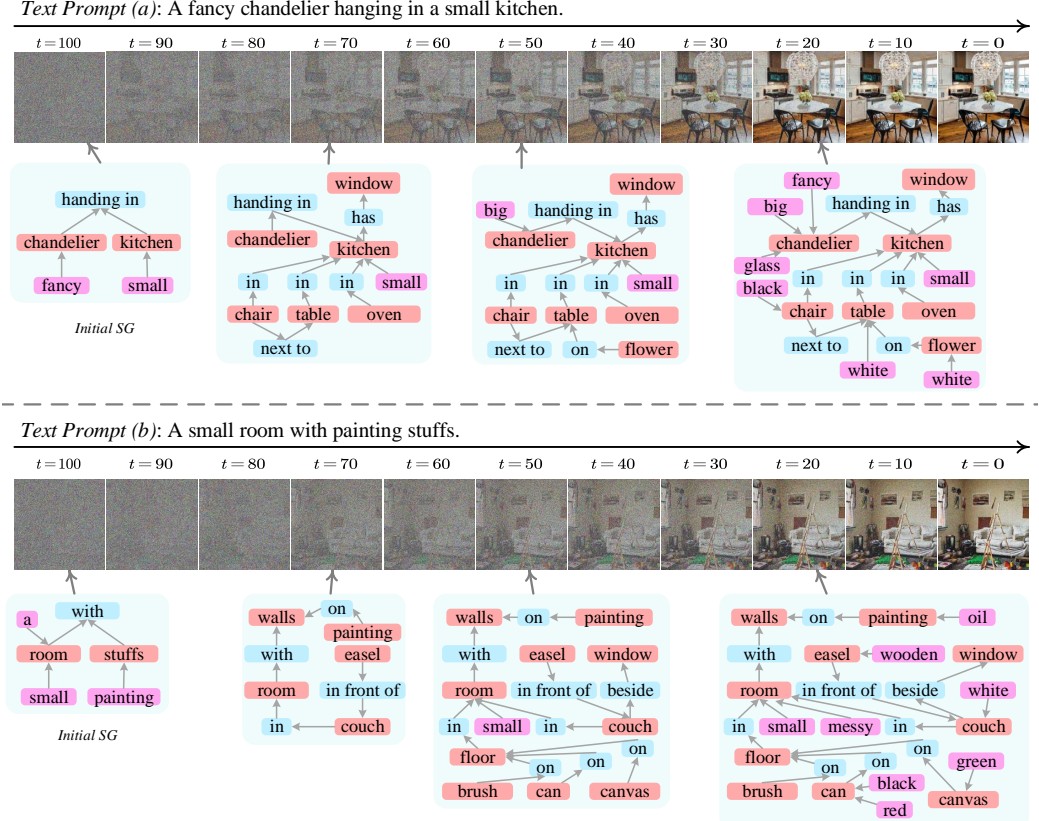

Figure 11: Visualization of T2I generation process at $t$=100, 90, $\cdots$, 0, along with the enriched SG at $t$=100, 70, 50, 20.

*Text Prompt:* Sparse narrow kitchen.

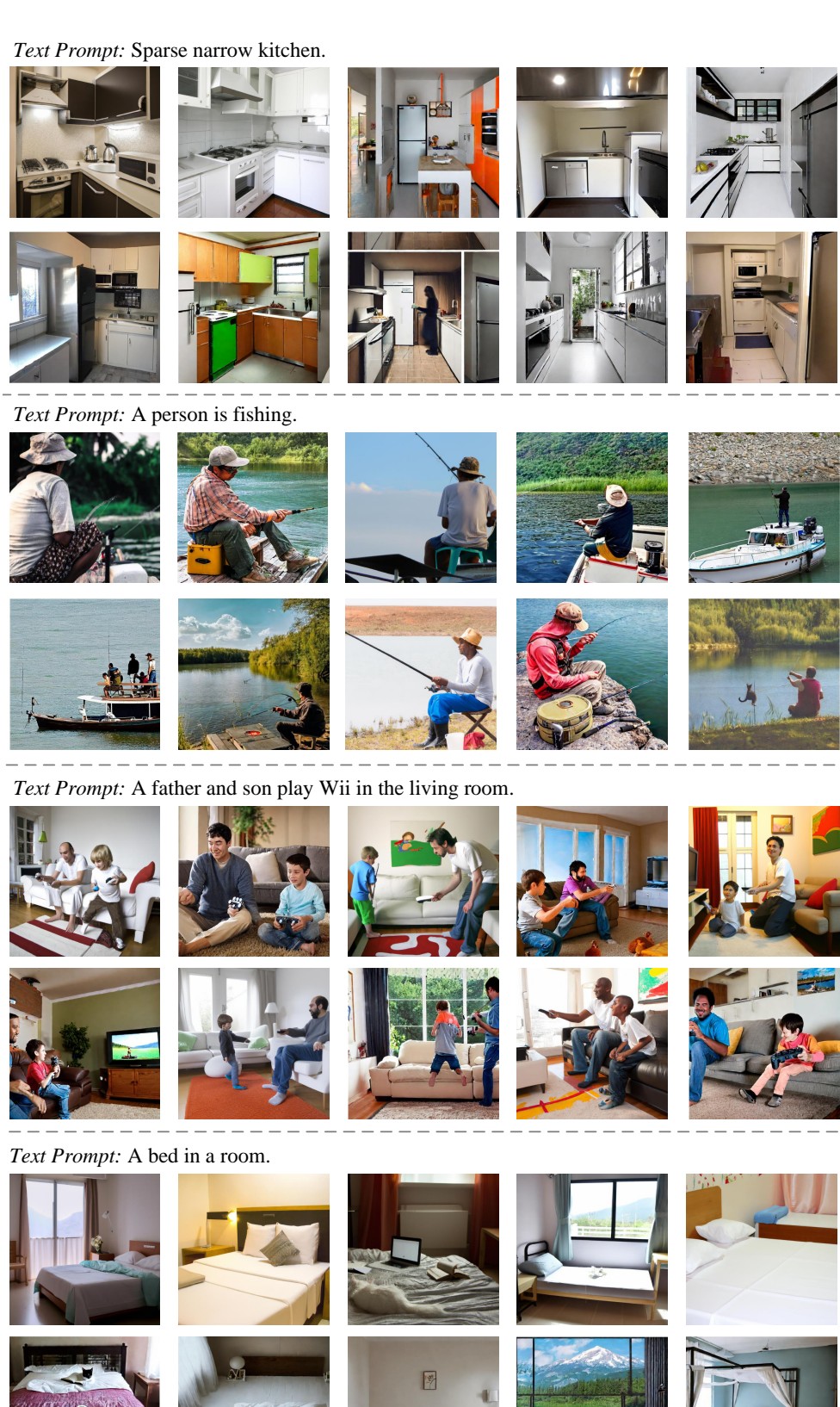

*Text Prompt:* A person is fishing.

*Text Prompt:* A father and son play Wii in the living room.

*Text Prompt:* A bed in a room.

Figure 12: More samples of abstract-to-intricate T2I generation via scene sampling mechanism.

*Text Prompt:* A person is baking.

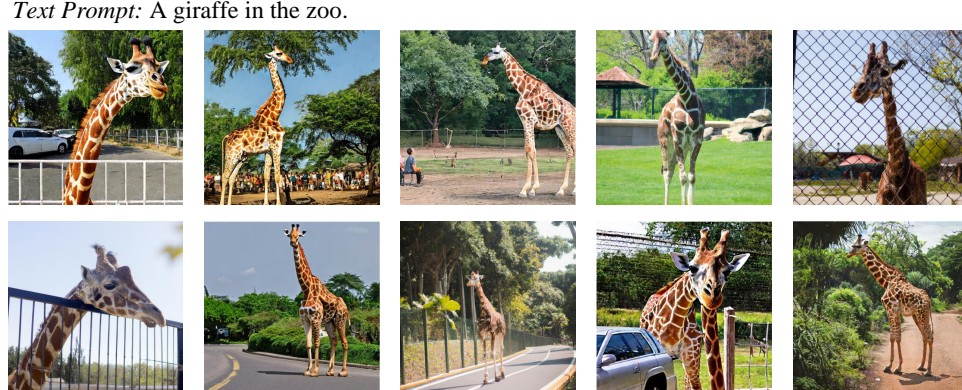

*Text Prompt:* A car and traffic light on a city street.

*Text Prompt:* A living room for painting.

*Text Prompt:* A giraffe in the zoo.

Figure 13: More generated samples by our model.

