# OpenReview forum: "Imagine That! Abstract-to-Intricate Text-to-Image Synthesis with Scene Graph Hallucination Diffusion"
_NeurIPS.cc/2023/Conference — NeurIPS 2023 poster_

### Official Review · Reviewer_GuSJ · 2023-06-16

**Soundness:** 2 fair
**Presentation:** 3 good
**Contribution:** 2 fair
**Rating:** 4
**Confidence:** 5

**Summary:**

In this paper, text-to-image synthesis under the abstract-to-intricate setting is studied. Firstly, the input prompt is hallucinated and expanded into feasible specific scene structures by the proposed SGH mechanism. Then, text-to-image synthesis is implemented through a diffusion-based synthesizer by gradually incorporating semantic scene structure induced from the SGH. Extensive experiments on COCO, especially on the abstract-to-intricate text-to-image setting, prove the method could contribute to synthesizing images reasonably and accurately under the simple text.

**Strengths:**

1.	The proposed abstract-to-intricate T2I is a vital research topic in the field of text-to-image synthesis since existing large models require delicate and well-designed prompts for controllable image synthesis.
2.	Abstract-to-intricate T2I is further promoted by the designed SGH mechanism, avoiding vision distraction from text and wrong focus enriched prompts.
3.	Extensive experiments including thorough metrics are conducted to prove the effectiveness. And corresponding analysis and discussion are reasonably stated.


**Weaknesses:**

1.	Existing experiments compared to the text-based enrichment methods are only conducted on manually selected dataset COCO-A2I with Place Norm and Progressive Verbs, which is not representative and convincing enough.
2.	There exists unfair comparison in T2I results on COCO, since the best FID and CLIP score of Frido are respectively 8.12 and 0.7915. And the best T2I baseline on COCO is not Frido. As far as I know, make-a-scene[1] achieves a 7.55 FID, which is not discussed.
3.	Scene graphs consist of two parts, which are node and edges representing semantics, bounding boxes referring to sizes and locations of objects. As proved by previous works, incorporating visual guidance into T2I training is beneficial. Why bounding boxes information is not included for training?
4.	Overclaims and inaccurate description in contributions:” We propose a diffusion-based model with SG guidances for highly controllable and scalable image generation.” Scene graph hallucinations might include unexpected concepts, which is not controlled by users.
5.	Unclear captions and inconsistent description. The caption of Fig.2 is unclear and lacks descriptions about each subfigure.
6.	Citations about scene graph-to-image synthesis and scene graph generation are not thoroughly included[2-4].
7.	Typos and inconsistent descriptions: In line 226, “diffusioninspired”. In line 245, Frido-G is inconsistent with the description on Tab.1.

[1] Gafni O, Polyak A, Ashual O, et al. Make-a-scene: Scene-based text-to-image generation with human priors[C]//Computer Vision–ECCV 2022: 17th European Conference, Tel Aviv, Israel, October 23–27, 2022, Proceedings, Part XV. Cham: Springer Nature Switzerland, 2022: 89-106.
[2] Sitong Su, Lianli Gao, Junchen Zhu, Jie Shao, and Jingkuan Song. 2021. Fully Functional Image Manipulation Using Scene Graphs in A Bounding-Box Free Way. Proceedings of the 29th ACM International Conference on Multimedia. Association for Computing Machinery, New York, NY, USA, 1784–1792. https://doi.org/10.1145/3474085.3475326
[3] Lyu X, Gao L, Guo Y, et al. Fine-grained predicates learning for scene graph generation[C]//Proceedings of the IEEE/CVF Conference on Computer Vision and Pattern Recognition. 2022: 19467-19475.
[4] Herzig R, Bar A, Xu H, et al. Learning canonical representations for scene graph to image generation[C]//Computer Vision–ECCV 2020: 16th European Conference, Glasgow, UK, August 23–28, 2020, Proceedings, Part XXVI 16. Springer International Publishing, 2020: 210-227.




**Questions:**

1.	Please provide results compared to text-based enrichment methods on COCO dataset.
2.	Please discuss results compared to the best Firdo other T2I baselines like make-a-scene.
3.	Why bounding boxes of scene graphs for visual guidance are not included in training? Will there be gain by adding bbox?
4.	Please modify writing issues and answer questions in 4,5,7 of Weakness.
5.	Citations are missing as described in 2, 6 of Weakness.


**Limitations:**

yes

---

> ### Author Rebuttal · Authors · 2023-08-09
>
> We sincerely thank you for your time and valuable comments. Your suggestions will surely help consolidate our paper. Following we present the point-to-point response to address your concerns. And if you feel our responses effectively relieve your concerns, please kindly reconsider your evaluation.
>
>  ---
> **Q1:  Existing experiments compared to the text-based enrichment methods are only conducted on manually selected dataset COCO-A2I with Place Norm and Progressive Verbs, which is not representative and convincing enough.  Please provide results compared to text-based enrichment methods on COCO dataset.**
>
> **A:** Here, we also provide the results by comparing them with the text-based enrichment methods on the overall COCO dataset, where pre-trained Frido is employed to generate images. Compared with the best baselines, the two enrichment approaches fail to exhibit superior results on the three evaluation metrics, indicating that text-based enrichment not only fails to ease the difficulty of existing models in image synthesis, but also causes the performance decrease of existing models. Moreover, these results further demonstrate the effectiveness of our proposed method.
> | Model | FID | IS | CLIP|
> | :-----: | :----: | :----: | :----: |
> |Frido|11.24|26.84|70.46|
> |SD-PG|15.78|24.76|65.74|
> |SPY|12.63|25.34|66.95|
>
>  ---
> **Q2:  There exists an unfair comparison in T2I results on COCO, since the best FID and CLIP score of Frido are respectively 8.12 and 0.7915. And the best T2I baseline on COCO is not Frido. As far as I know, make-a-scene[1] achieves a 7.55 FID, which is not discussed.**
>
> **A:**  Sorry for the mistake of not providing the full comparison with Frido. We conduct a re-evaluation using the open-source checkpoint provided by the authors and the same evaluation code. Following are the experimental results. Comparatively, our model outperforms Frido-CLIP and exhibits superior performance, particularly on the COCO-A2I dataset, showcasing significant improvements. This again validates the effectiveness of the proposed method.
>
> Result on COCO:
> |Model|FID|IS|CLIP|
> | :-----: | :----: | :----: | :----: |
> |Ours|10.19|29.96|74.83|
> |Frido-CLIP|10.87|27.30|73.46|
>
> Result on COCO-A2I:
> |Model|FID|IS|CLIP|
> | :-----: | :----: | :----: | :----: |
> |Ours|31.25|28.63|71.29|
> |Frido-CLIP|38.45|19.22|69.20|
>
> As for the Make-A-Scene model you indicated here, it is actually trained on the total collection of **CC12m**, **CC**, **YFCC100m**, **Redcaps**, and **COCO**, in amount to 35m text-image pairs, a definitely much larger volume data than ours, which enables to achieve a much lower FID score. Therefore, the direct comparisons with Make-A-Scene can be unfair. Also, the evaluation on Make-A-Scene could be unfeasible for us, as it does not offer an open-source checkpoint.
>
> ---
> **Q3:  Scene graphs consist of two parts, which are nodes and edges representing semantics, bounding boxes referring to sizes and locations of objects. As proved by previous works, incorporating visual guidance into T2I training is beneficial. Why bounding boxes information is not included for training?**
>
> **A:** In this work, we placed the major emphasis on modeling the object nodes and semantic relationships between objects and attributes. The experimental results demonstrate that just employing such information has notably improved the quality of image generation.
>
> That being said, we totally agree that the incorporation of layout information (e.g., the bounding box) will be promising to gain more controllable and high-quality image generation [1], which we consider as future work.
>
> [1] LayoutGPT: Compositional Visual Planning and Generation with Large Language Models
>
>  ---
> **Q4:  Overclaims and inaccurate description in contributions:” We propose a diffusion-based model with SG guidances for highly controllable and scalable image generation.” Scene graph hallucinations might include unexpected concepts, which is not controlled by users.**
>
> **A:** Actually, here we emphasize the **controllable generation** more about the highly structured SG representations (i.e., the 'object-predicate-subject' triplets) that exhibit the ability of precise control over the image generation process.
>
> Your point could also be reasonable to a certain extent that, hallucinations may come with unexpected outcomes. But in our work, by training with sufficient volume of supervised data, our SGH module learns to (more tends to) have valid imagination (as we evaluated and answered this question in our _Q2_ of section 4.4, line 304). So this is quite different from the hallucination by the existing LLMs; they are two separate concepts. But, we will improve our expressions and make it clearer and more accurate on this.
>
> ---
> **Q5:  Unclear captions and inconsistent description. The caption of Fig.2 is unclear and lacks descriptions about each subfigure.  Typos and inconsistent descriptions: In line 226, “diffusioninspired”. In line 245, Frido-G is inconsistent with the description on Tab.1**
>
> **A:** Thanks for going through the paper this carefully; we will correct them all in the revision.
>
> ---
> **Q6: Citations about scene graph-to-image synthesis and scene graph generation are not thoroughly included.**
>
> **A:** Thanks for indicating the missing literature, we will carefully check them and consider including them into the related work.

---

> > ### Comment · Reviewer_GuSJ · 2023-08-21
> >
> > Thanks for your response.
> > This rebuttal addressed some of my concerns.
> > However, there are lots of OVERCLAIMED statements in this paper.
> > For example, 'We are the first to study the novel T2I setup of intricate image synthesis from succinct abstract texts,'
> > Actually, this kind of setup can be easily modified from existing T2I tasks.
> > I will keep my rating.

---

> > > ### Author Response · Authors · 2023-08-21
> > >
> > > Dear Reviewer#GuSJ,
> > >
> > > Thanks for your feedback. We'd like to make further clarification to your point.
> > >
> > > While the T2I model could be easily adopted from existing work directly for the abstract-to-intricate (A2I) T2I task as you stated, it may largely fail this kind of A2I setting surely, which has been extensively evaluated in our experiments.
> > > As A2I T2I has not been studied before, this makes it the critical focus of this work.
> > > That is, we have already dedicated significant effort to addressing this matter, proposing an efficient model that we substantiated through comprehensive experiments.
> > > Thus, it is no doubt we are the pioneers in resolving this problem.
> > >
> > > Besides, would you mind specifying which part of the concerns remain unaddressed in our response? We would be more than willing to offer additional clarification.
> > >
> > > Best.

---

> ### Author Response · Authors · 2023-08-17
>
> Dear Reviewer#GuSJ,
>
> We would like to thank you again for your efforts and valuable feedback.
> Your comments are essential to help us improve the quality of our work.
> To address your main concerns about the rationale on COCO-A2I data construction and unfair comparison, we during rebuttal have run the experiments under a more reasonable setting by your suggestion, and presented the updated results.
> We kindly hope that you can take some time to check our response and re-evaluate our paper based on our replies.
> If you have any further concerns or questions, please do not hesitate to let us know. We will be happy to address them promptly.
>
> Best Regards.

---

### Official Review · Reviewer_QdRJ · 2023-06-29

**Soundness:** 3 good
**Presentation:** 3 good
**Contribution:** 3 good
**Rating:** 7
**Confidence:** 3

**Summary:**

This paper studies a new setup of generating intricate images from abstract prompts. To overcome the issues of vision distraction and wrong binding of using text as the condition to generate images, the author proposes a two stage pipeline by first generating scene graph from abstract text inputs, and then condition on the sythesized scene graph, another model generates images. The proposed diffusion model with SG guidance showcases its controllability and interoperability and it achieves new SoTA results in the abstract-to-intricate T2I setup. Overall, the paper is well-written and the proposed method is interesting and novel.


**Strengths:**

The introduction section is easy to follow, they provide examples to show the issues of the current T2I generation and motivate the scene graph representation as a potential solution.

The proposed SGH model is interesting and well adopted from VQ-diffusion etc for image modelling to the problem of scene graph modelling.

The experiments are convincing with strong performance on COCO, outperforming competitive baselines LDM, VQ-diffusion and Frido, they also construct an abstract-to-intricate dataset from COCO and demonstrate SoTA performance in this setup.


**Weaknesses:**

One claim of the paper is that SG guidance helps image generation with strong semantic controllability, it is not clear to me which experiments can support this claim.

In table 3, it seems like replacing the HSI module with GCN encoding only drops a little bit of the performance, it is questionable if the design of HSI is necessary.


**Questions:**

It seems like using SG as condition/guidance will lower the FID compared to text conditioning and also in qualitative results, details like faces will be blurry. Any idea how to improve this and explanation of the cause?

Any idea after enriching the SG, if users want to modify some nodes, how would the user interact with the SG/model?


**Limitations:**

As pointed out by the author, the proposed method depends on the generation quality of the SG, while a large-scale SG dataset is rare. Plus, since the pipeline needs to modify the conditioning of the LDM, how to better leverage pre-trained large T2I model is worth investigating.

---

> ### Author Rebuttal · Authors · 2023-08-09
>
> We sincerely thank you that you acknowledge the strengths of our work. Your support is the source of the power to push us forward and further enhance the paper. Following, we show the response to your questions one by one.
>
> ---
> **Q1:  One claim of the paper is that SG guidance helps image generation with strong semantic controllability, it is not clear to me which experiments can support this claim.**
>
> **A:** Actually, we have done the evaluation for this claim, please kindly refer to _Q1_ in section 4.5 (line 288). To evaluate the semantic controllability of SGs in image generation, we make comparisons with baselines (i.e., LDM, Frido) without SG guidance by measuring the semantic similarity (measured by _CLIP_ score) and semantic structural alignment (measured by _TriRec._ metric). As shown in Figure 6, the system with the guidance of SG consistently performs much better than the baselines in terms of those two metrics, indicating that SG guidance enhances the semantic controllability of the T2I process, and leads to high-faithful generation.
>
> ---
> **Q2:  In Table 3, it seems like replacing the HSI module with GCN encoding only drops a little bit of the performance, it is questionable if the design of HSI is necessary.**
>
> **A:** It is our mistake in wrongly calculating the numbers in Table 3. We incorrectly note the performance drop as 1.17. But the actual performance drop should be 4.71(=35.96-31.25) on the COCO-A2I dataset when replacing the HSI module with GCN. This notable decrease actually underscores the indispensability of the HSI module.
>
> Thanks for pointing this out. We will correct it.
>
> ---
> **Q3:  It seems like using SG as condition/guidance will lower the FID compared to text conditioning and also in qualitative results, details like faces will be blurry. Any idea how to improve this and an explanation of the cause?**
>
> **A:** FID measures the Fréchet Distance between the distribution of the synthetic images and real-world images within the feature space. In light of this, the lower FID score corresponds to the generation of more realistic images. Therefore, the model additionally using SG as condition/guidance achieves a lower FID score compared to the model only conditioned on text, demonstrating the former model excels in better image generation.
>
> As seen from the qualitative results, compared to the baselines, our proposed model has significantly enhanced the quality of face synthesis than the baselines. Besides, we kindly note that the images have been compressed to some extent when compiling into PDF file, resulting in blurriness and a decrease in resolution. Later upon acceptance, we will show the high-resolution resulting images on our project page to the public.
>
> ---
> **Q4:  Any idea after enriching the SG, if users want to modify some nodes, how would the user interact with the SG/model?**
>
> **A:** Good idea, which seems very promising in our setting. Actually using SG guidance for diffusion-based image editing by the user is feasible, because SG is a highly structural representation that enables more interactive manipulation for users and more controllable image generation.
>
> In our designed framework, the idea can be implemented by taking the user-modified SG as the conditions and then employing the hierarchical scene integration module to directly fuse the SG feature representations into the diffusion model without SG hallucination. However, in our current system, we did not cover this point. Thanks for your output, we will mention this part as future work.
>
> ---
> **Q5: As pointed out by the author, the proposed method depends on the generation quality of the SG, while a large-scale SG dataset is rare. Plus, since the pipeline needs to modify the conditioning of the LDM, how to better leverage the pre-trained large T2I model is worth investigating.**
>
> **A:** Thank you for your insights. Indeed, the size of the SG dataset plays a pivotal role in the imaging ability of the proposed scene graph hallucination (SGH) module. Actually, we have leveraged the rich number of annotated Visual Genome (VG) dataset (with 62k annotated images) for pre-training the SGH module, and then we fine-tune the SGH on the COCO dataset (83k). We believe the data quantity of the training SGH module can be quite substantial in size. Based on the experiments, the proposed SGH moudel is capable of imaging reasonable SGs and facilitates the generation of high-quality images. Of course, it is well believed that with more data, the performance of SGH can be further boosted, for producing more accurate and visually-enriched imagination.

---

> > ### Comment · Reviewer_QdRJ · 2023-08-21
> >
> > Thank the author for the response. I retain my original rating and encourage authors to open-source the code for the research community.

---

> > > ### Author Response · Authors · 2023-08-21
> > >
> > > Dear Reviewer #QdRJ,
> > >
> > > Thanks again for your acknowledgment. Sure, we will release the codes and resources upon acceptance.
> > >
> > > Best.

---

### Official Review · Reviewer_mzvT · 2023-07-01

**Soundness:** 4 excellent
**Presentation:** 4 excellent
**Contribution:** 4 excellent
**Rating:** 10
**Confidence:** 5

**Summary:**

The paper proposes a novel approach to text-to-image (T2I) synthesis, specifically focusing on generating intricate visual content from simple abstract text prompts, aka., abstract-to-intricate (A2I) setting. The proposed mechanism, named scene-graph hallucination (SGH), expands the initial scene graph (SG) of the input prompt with more feasible specific scene structures using discrete diffusion technique. A continuous-state diffusion model then serves as the T2I synthesizer navigated by the semantic scene structure induced from the SGH module. Additionally, this paper further devises a scene sampling mechanism to generate various scene graphs. They also construct a more challenging benchmark data, called COCO-A2I, to effectively evaluate the models under the abstract-to-intricate setting. Experiments on two benchmark datasets shows that leveraging SG imagination helps better image generation, where Salad could hallucinate certain scene clues to facilitate the abstract-to-intricate T2I generation. The study also contributes to a better understanding of the efficacy and rationale behind scene graph features, with potential applications in other tasks such as image editing and text-to-video generation.

**Strengths:**

Overall, I enjoy the work much where it studies the specific conditioned image synthesis from an interesting and realistic perspective, with robust and novel technical methods. The paper seems very solid to me, with thorough evaluations from many angles, and the details are given quite sufficiently (with long and informative appendix content). While this is a very technical paper, there is immense interest in diffusion models under such novel perspective. I believe this paper will have the potential to unlock interesting future research.

**Interesting and meaningful perspective.**
The paper studies how to generate intricate images from succinct abstract prompts. This can be a very interesting and realistic perspective in many scenarios. The issue essentially lies in the natural modality asymmetry between language and vision, where strict one-to-one correspondences do not exist. This becomes particularly challenging when T2I systems attempt to capture the nuanced content of the prompts and generate corresponding high-quality images, thereby highlighting the need for deep semantic understanding in these systems.

**Innovative methodology.**
The authors propose a novel approach to the task of abstract-to-intricate T2I synthesis, where the image-generating process is controlled and navigated by the underlying semantic scene structure. On the one hand, it satisfies the human intuition for handling this task, which is pretty interesting and makes a lot sense to me. If the system could hallucinate some concrete textual clues which has a more corresponding relationship to the visual scenes, the generation process will ease largely. Technically, the devised method, called scene-graph hallucination (SGH), expands the initial scene graph (SG) of the input prompt by iteratively evolving new scene elements via discrete diffusion model, which are theoretically sound and empirically validated through experimental results. On the other hand, the hierarchical scene integration mechanism is able to ensure the highly effective guidance of the semantic scene features.

**Solid evaluations and convincing analyses.**
The results are presented clearly and concisely, showcasing a significant improvement for the abstract-to-intricate T2I task at hand. Some in-depth analyses are provided to offer substantial evidence and valuable insights from all different perspectives to support the claims made, such as, exploring the effectiveness and rationale behind the employed scene graph features. Also, the comprehensive implementation details will enhance reproducibility and facilitates the smooth adoption of the proposed approach. The codes are provided.

**Well-structured presentation and clear writing.**
This paper is exceptionally clear, well-written (except some notation problems) and good illustration, making it easy to understand. The details are given quite sufficiently, with long and informative appendix content. I’ve gone through the appendix and find almost everything I wanted to see.

**Weaknesses:**

While I think the paper is solid, it can be significantly improved further if the following issues (major or minor) are considered properly:

(ordered by appearance in the paper)

1. In Figure 1, the issues of vision distraction and wrong binding appear to represent the same issue where the resulted images deviate from the original user intention. Besides, it would be beneficial to maintain consistent terminology throughout. For example, in Figure 1(a), it is referred to as 'vision distraction', whereas in the article, it is mentioned as "visual distraction."

2. Although the authors devise a scene sampling mechanism to generate various scene graphs for diversified image syntheses during inference, the diversity of scene graph remains inadequate. This could be due to the limited number of categories of objects, attributes or relationships in the SGH module, which restricts the model's imagination and exploration capacity. In real scenarios, the objects, attributes can be quite multifarious, for example, the color of the T-shirts.

3. Potentially lack of comparison of the existing scene graph completion work. The performance of scene graph hallucination has a huge impact on the performance of abstract-to-intricate T2I task. It would be interesting to see the performance comparison on existing work about scene graph hallucination, such as [1,2]. Moreover, another intuitive method is the pipeline method, i.e., first scene graph hallucination and then scene graph generation based the imagined SG. Therefore, some evaluating experiments should be provided to make a comparison with the pipeline method.

4. The authors may possibly overlook the imagination ability of the T2I model itself. For example, existing T2I models can synthesis an image based on the prompt, 'a man'. Is the appearance of the man in the generated image, such as hair color, attributed to the imagination of the T2I model?  In other words, the authors fail to provide a clear definition of when the model needs to image. Furthermore, they do not delve into what scene graph needs to be detailed in order to generate intricate images effectively. For instance, in Figure 1(b), the enrich SG not extremely correspond to the image as the screen on the wall are not demonstrated in the enriched SG, but the model is still able to generate it.

5. Some unclear and confusing annotations:

- In Figure 2 right bottom, it is confusing $a_{n, 1}$ whether one value, i.e., the m-th attribute of the object $o_1$, or many values. Same issues for $r_{n,k}$.

- In line 150, the distribution should be $p_{\theta}(s_t|s_{t+1}, y)$.

- Is $\mathcal{B}{s_t}$ a row one-hot vector or column one-hot vector? This should be clarified. Besides, there should be a transition symbol in Eq. 2.

- In Eq. 1, the $\mathcal{L}_{vlb}$ is not explicitly designed for the optimization of conditional image generation. Thus, some clarification should be added, and the correct loss function is demonstrated in Appendix B.1, specifically Equations 19-22.

- No illustration of the $d$ in Eq. 7.

- In the Implementation section, the version of CLIP mentioned is not consistent with the information provided in Appendix C.5.

- No demonstration of the hyper-parameters? top-A and the temperature $\eta$ in the inference?

- In Table 3, the NTD-CA should be Eq. 5, not Eq. 17

- In Line 314, not ‘object-object’ pairs but ‘subject-predicate-object’ triplets.

- In Line 738, ‘2.005’ should be written as ‘2,005’.

- In Table 5, the second ‘Max’ should be ‘Avg.’


6. The paper lacks in-depth analysis of the circumstances under which the proposed SGH module fails to generate reasonable scene graphs, leaving a gap in understanding its limitations.

7. In the section 4.5, the question Q2 is interesting but strikes me as a rather weird way to answer the question. The Figure 7 precisely answers that the SGH is able to induce intricate SGs. As discussed in Q1, there is a strong semantic alignment between the input prompts and the generated images guided of the SG. Consequently, it becomes relatively straightforward to achieve a high TriRec. Score by comparing the induced SG with the SG of the generated image. Nonetheless, this response lacks directness and comprehensiveness regarding whether SGH is capable of producing reasonable SGs. Besides, the concept of a "reasonable" SG remains ambiguous, For example, if the ‘table-in-room’ is a reasonable scene triplet, what about the triplet ‘table-in-ocean’? If a reasonable SG means some SGs that conforms natural conditions, can our proposed model generate some abstract or unconventional images?


[1] Garg S, Dhamo H, Farshad A, et al. Unconditional scene graph generation. 2021.

[2] Agarwal R, Chandra T S, Patil V, et al. GEMS: Scene Expansion using Generative Models of Graphs. 2023

**Questions:**

1. Is there any further explanation about the order of the Text-CA, Graph-CA, NTD-CA?
2. During the SG hallucination, other visual scenes are also introduced. How does the proposed method deal with the visual distraction issues?
3. How to ensure that the objects, attributes, and relationships in the initial SGs are always remain in the imaged SGs?
4. Has the main results in Table 1&2 adopted the scene sampling mechanism? So is this the average of multiple experimental results?
5. In HIS, which semantic level has a significant impact on task performance?
6. How to leverage the VG dataset to train the SGH module?

**Limitations:**

I do not foresee any potential negative impact from this work.

---

> ### Author Rebuttal · Authors · 2023-08-09
>
> We appreciate it so much that you went through our work so deep. And we are very much excited to receive your such strong support, which will definitely push us moving forward. Following we will answer your questions one by one.
>
> ---
> **Q1:  In Figure 1, the issues of vision distraction and wrong binding appear to represent …**
>
> **A:** Our aim is to analyze this phenomenon at a detailed level, as while the problems in the generated images are alike, there are nuanced variations in their underlying causes. We appreciate your careful review, and in revision, we will ensure the consistent use of terminology.
>
> ---
> **Q2: Although the authors devise a scene sampling mechanism to …, the diversity of scene graph remains inadequate.**
>
> **A:** Based on the current experimental results, we see that the imagination module has good imagination capacity, which can help generate intricate images from abstract prompts. However, we also acknowledge that the model's imagination and exploration capabilities are limited. As part of our future work, we will explore how to alleviate this limitation.
>
> ---
> **Q3: lack of comparison… Moreover, experiments should…**
>
> **A:** Following [2], we compute the Maximum Mean Discrepancy (MMD) for node and edge types, as well as sub-graph similarities (referred to as NSPDK*) to compare our method with baselines in scene graph expansion. As shown in the table, our method consistently outperforms the baselines, indicating graphs generated by our model are more meaningful and better align with  the observed SG distribution.
>  |Matrix|GraphRNN|SceneGraphGen|GEMS|Ours|
> | :-----: | :----: | :----: | :----: | :----: |
> | Node ($\times 10^4 \downarrow$ )|5.44|5.92|5.19|4.85|
> | Edge ($\times 10^2 \downarrow$)|22.38|0.83|1.13|0.63|
> | NSPDK* ($\times 10^2 \downarrow$)|22.60|0.73|1.21|0.69|
>
> ---
> **Q4:  The authors may possibly overlook the imagination ability of the T2I model itself …**
>
> **A:** As discussed in Q2, our work lies in enhancing generative models' capacity to comprehend abstract text by envisioning key objects that concretize the abstract text, rather than exhaustively generating all image details. Additionally, we take the imagined SGs as skeletons, which means the SGs and images are not in one-to-one correspondence. Consequently, it is possible that the enriched scene graph does not precisely match the generated image.
>
> ---
> **Q5:  Some unclear and confusing annotations.**
>
> **A:** Thank you so much for the careful reviewing, we will revise them and further carefully double-check all the content.
>
> ---
> **Q6:  The paper lacks an in-depth analysis … leaving a gap in understanding its limitations.**
>
> **A:** Because of dataset constraints, our method encounters difficulties with infrequent abstract words that prove challenging to concretize using specific key objects, like 'yearn' and 'freedom' in the prompt 'A person who yearns for freedom'. resulting in the failure of high-quality image generation. Actually, we outline a potential vulnerability assessment in Appendix A.3, please refer to it.
>
> ---
> **Q7: In section 4.5, the question Q2 is interesting but strikes me as a rather weird way to answer the question…**
>
> **A:** Firstly, we assume that the SGs of gold images entail reasonable scenes. Therefore, if the SGs induced by SGH exhibit a high alignment with the SGs of the gold images, i.e., achieving a high TriRec. score, it indicates the capability of SGH to induce reasonable SGs, directly addressing the question Q2.
>
> Secondly, our proposed method, due to the scene sampling strategy, has the potential to generate unconventional images. This is because we take the TOP-A category candidates and perform sampling over these candidates instead of always picking the best prediction of the category of any node during inference.
>
> ---
> **Q8: Is there any further explanation about the order of the Text-CA, Graph-CA, NTD-CA?**
>
> **A:** In the experiment, we also devise the other orders between three condition injection modules. Based on the experiments, we observe that the order illustrated in Figure 2 shows the best performance for SG hallucination.
>
> ---
> **Q9:  During the SG hallucination… How does the proposed method deal with visual distraction issues?**
>
> **A:** In this work, there are two approaches to address visual distraction issues. The one is rational SG imagination, aiming to align the imagined SGs with the gold images while minimizing unwanted information induced. The other one is controllable image generation, where the image generation is fine-grainedly guided by the objects, attributes, and relationships specified in the SGs.
>
> ---
> **Q10:  How to ensure that the objects … remain in the imaged SGs?**
>
> **A:** Indeed, it is not essential to ensure a strict correspondence between the initial scene graphs (SGs) and the imaged SGs, since the imaged SGs involves many more concrete objects, while the initial SGs maybe include certain abstract objects.
>
> ---
> **Q11:  Have the main results in Table 1&2 adopted the scene sampling mechanism? So is this the average of multiple experimental results?**
>
> **A:** Yes, it is. We present the average of multiple experimental results.
>
> ---
> **Q12:  In HIS, which semantic level has a significant impact on task performance?**
>
> **A:** The information at different semantic levels contributes to the task performances to different extent, with the semantic information at the relation level showing a notable influence on the task. This is attributed to the fact that the information at relation level encompasses both object-level details and higher-level information.
>
> ---
> **Q13:  How to leverage the VG dataset to train the SGH module?**
>
> **A:** As no captions for images in the VG dataset, we first extract the seed graph as initial graph by the algorithm in [1]. Then, we transform the seed graph into a token sequence as the text prompt. Finally, we use the VG data to train our SGH module.
>
> [1] GEMS: Scene Expansion using Generative Models of Graphs. WACV 2023

---

> > ### Comment · Reviewer_mzvT · 2023-08-12
> > **Response to author rebuttal**
> >
> > Thank the authors for the detailed response, which has effectively addressed most of my concerns.
> > But I still have a few lingering points to discuss:
> >
> > 1. I remain somewhat perplexed about how the proposed model determines what input prompts are needed to engage in the imagination process.
> >
> > 2. I've also glanced at the comments by other reviewers. With regard to the task definition, I find Reviewer DqEY's perspective quite compelling. Text-to-image generation inherently involves a complex transition from abstract concepts to intricate visualizations due to the inherent asymmetry between language and vision. Why does this work solely concentrate on place nouns and progressive verbs? Put differently, there seems to be a lack of a comprehensive definition of abstract terms. If this is caused by the COCO data distribution, like the place nouns and progressive verbs are the most common cases, then my suggestion could be, authors are encouraged to further enrich the proposed COCO-A2I data, maybe say XXX-A2I, by including and collecting more scenarios.
> >
> > 3. The capacity for imagination within the proposed method appears to be limited by the generalization and diversity of the training dataset, right? Could the authors provide more insight into the potential strategies available to enhance imaginative capabilities while concurrently mitigating the risk of hallucination?
> >
> > Overall, I consider this paper to be of great value. The addressed problem is intriguing and interesting, and under the conditions of a well-conducted experiment, the proposed method demonstrates a marked superiority over the existing method. Therefore, I will maintain my earlier assessment.

---

> > > ### Author Response · Authors · 2023-08-13
> > >
> > > Thank you so much for the prompt responses and high recognition of our work. Following we will show the response to your questions one by one.
> > >
> > > ***
> > >
> > >
> > > **Q1: I remain somewhat perplexed about how the proposed model determines what input prompts are needed to engage in the imagination process.**
> > >
> > > **A:** During the training, we utilize the scene graphs derived from the generated images as supervised data to optimize the imagination model.
> > > In essence, when discrepancies arise between the scene graph associated with the input prompts and the desired scene graph, the model endeavors to enhance its ability to envision the input prompts, subsequently expanding it into a scene graph that harmonizes more cohesively with the semantic scene structure of the target images. Notably, since a pronounced discrepancy exists between the abstract input and the scene graph of the target image, the model allocates greater focus to mastering the imagination for abstract words.
> > >
> > >
> > > ***
> > >
> > > **Q2: I've also glanced at the comments by other reviewers. With regard to the task definition, I find Reviewer DqEY's perspective quite compelling. Text-to-image generation inherently involves a complex transition from abstract concepts to intricate visualizations due to the inherent asymmetry between language and vision. Why does this work solely concentrate on place nouns and progressive verbs? Put differently, there seems to be a lack of a comprehensive definition of abstract terms. If this is caused by the COCO data distribution, like the place nouns and progressive verbs are the most common cases, then my suggestion could be, authors are encouraged to further enrich the proposed COCO-A2I data, maybe say XXX-A2I, by including and collecting more scenarios.**
> > >
> > > **A:** Thanks for the reviewer's valuable and constructive suggestions. Actually, we already embark on the process of enriching the A2I data by collecting more data through the following approaches:
> > > 1) we leverage the existing image-caption pairs to collect more instances under various scenarios, such as [CC12M](https://github.com/google-research-datasets/conceptual-12m), or [CommonPool](https://arxiv.org/abs/2304.14108);
> > > 2) we construct various abstract input prompts by designing different templates within filling with different types of abstract words.
> > >
> > >
> > > ***
> > >
> > > **Q3: The capacity for imagination within the proposed method appears to be limited by the generalization and diversity of the training dataset, right? Could the authors provide more insight into the potential strategies available to enhance imaginative capabilities while concurrently mitigating the risk of hallucination?**
> > >
> > > **A:** We believe there are promising approaches to enhance the capacity for the imagination of the proposed method.
> > > Firstly, training model with more diverse datasets. We can leverage existing large-scale datasets to further optimize the scene graph imagination module, enabling the model to perform scene imagination for a wider range of abstract content.
> > > Secondly, extending the depth of imagination. This involves expanding the categories of objects, attributes, and relationships within scene graphs that the model can currently image. This refinement will empower the model to envision a broader array of intricate scene components.
> > > Thirdly, integrating with LLMs. We can explore LLM's potential to comprehend and expand scene graphs as LLMs have shown multifaced abilities in various tasks. Meanwhile, we can leverage the retrieved knowledge from Wikipedia or other authoritative platforms to mitigate the hallucination issues in LLM.
> > >
> > >
> > >
> > > ***
> > >
> > >
> > > Thanks again for your interactions. And if you have further inquiries, please do not hesitate to reply to this thread.

---

> > > > ### Comment · Reviewer_mzvT · 2023-08-14
> > > > **Maintain my initial rating**
> > > >
> > > > Thanks. The authors solve all my concerns. In general, I believe this work has substantial value as stated in my initial judgment

---

### Official Review · Reviewer_DqEY · 2023-07-07

**Soundness:** 2 fair
**Presentation:** 2 fair
**Contribution:** 3 good
**Rating:** 5
**Confidence:** 4

**Summary:**

The paper proposes a new setting (or sub-domain) of text-to-image generation (T2I), namely abstract-to-intricate T2I. To tackle the new setting, the authors propose a method (Salad) to enrich the scene graphs parsed from the text prompts based on the discrete diffusion models. The enriched scene graphs are used as guidance to generate complicated scenes that align better with the initial concise prompts. They report quantitative results and analysis experiments to show the effectiveness of the two-stage system in multiple metrics.

**Strengths:**

- The paper addresses a practical and important problem. Users of the T2I system, in reality, have to write unreasonably long text prompts to generate images with plausible scenes/styles/semantics. Abstract-to-intricate T2I could benefit the users with a more faithful and efficient generation process.

- The scene graph hallucination stage achieved with the discrete diffusion model is interesting. Enriching the text prompt in the scene graph space seems like a more controllable and stable process that can preserve the faithfulness of relations and attributes.

- The authors conduct an extensive analysis of the components of the system to demonstrate its effectiveness.

**Weaknesses:**

- Task definition. While I understand the concept of abstract-to-intricate, I think the work lacks a more rigorous definition of the task. It seems unclear why places and progressive verbs are included and other nouns or verbs are discarded in the COCO-A2I. I tried the failure prompt in Fig. 1 on Stable Diffusion, and I got good results most of the time. So what makes these prompts an A2I problem instead of a faithfulness problem of all T2I models?

- My major concern is about the experimental setup. It seems that the Salad system is using the pre-trained Stable Diffusion as the image generator compared to other methods like Frido and LDM-G. The comparison could be unfair as all these baselines are trained with much fewer images and inherently have higher FID scores on MSCOCO. In addition, the authors use Frido for text-based enrichment approaches.

- The writing needs improvement. There are multiple typos throughout the paper and some inaccurate sentences. For instance, line 133 "attribute (o)", line 226 "baselines: stable diffusioninspired by [6].", line 235-236 "For the SIS module, we load the parameters of Stable Diffusion3 (v1.4) as the initialization.", line 238 "UNet" (UNet in Stable Diffusion?).

**Questions:**

- What is the SGH module's output and the SIS step's input? Fig. 2 seems like you are inputting the scene graph matrix into the UNet, while Sec. 3.2 and Fig. 3 show that you convert the graphs into prompts and feed them to the attention layers.

- Did you fine-tune the Stable Diffusion model by saying "For the SIS module, we load the parameters of Stable Diffusion3 (v1.4) as the initialization."?

- Why do all models achieve such high CLIP scores? Is the CLIP score the cosine similarity between image-text pairs or the CLIP R-precision metric?

---

> ### Author Rebuttal · Authors · 2023-08-09
>
> We sincerely thank you for going through our paper carefully, and providing valuable constructive feedback, which will surely benefit our work. Following we will address your concerns. And we sincerely hope you can raise your evaluation when you feel that we relieve your concerns.
>
> ---
> **Q1: Task definition. I think the work lacks a more rigorous definition of the task. It seems unclear why places and progressive verbs are included and other nouns or verbs are discarded in the COCO-A2I. I tried the failure prompt in Fig. 1 on Stable Diffusion, and I got good results. So what makes these prompts an A2I problem instead of a faithfulness problem of all T2I models?**
>
> **A:** Thank you for pointing this out. Here we can give a rigorous definition of the setting. This work is dedicated to improving the quality of intricate image generation conditioned on succinct abstract prompts, where the prompts are concise and highly abstract and often encapsulate complex scenes. Actually, abstract words have many categories, such as _place noun_, _progressive verb_, _social events_ (e.g., party, concert, conference), and _experience nouns_ (e.g., adventure, journey, festival). For the COCO data, we conduct the pilot study and found that the two types of abstract words, _place nouns_ and _progressive verbs_, are the most common, with a proportion of up to 95% and other types of abstract words account for a very small portion. Thus, we mainly selected these two types for COCO-A2I dataset.
>
> The results in Fig1 were obtained by the Stable Diffusion (v1.4) checkpoint at the time we were doing our project. Currently, the SD model has been updated several times as it trained on more datasets, achieving better results. Therefore, if you try it now or recently, you may have a chance to obtain good results. That being said, there is no guarantee of good performance for using SD on those abstract prompts. To gain an empirical result, we randomly select 50 abstract prompts from the COCO-A2I dataset, and input them into the latest Stable Diffusion (v2.1), with only a success rate of generating satisfactory results of 54%. Therefore, enabling T2I generative models with imagination ability is still non-trivial. At the same time, we acknowledge that the A2I problem is inherently a problem of addressing the faithfulness of T2I models.
>
> ---
> **Q2:  My major concern is about the experimental setup. Salad system is using the pre-trained Stable Diffusion as the image generator compared to other methods like Frido and LDM-G. The comparison could be unfair as all these baselines are trained with much fewer images. In addition, the authors use Frido for text-based enrichment approaches.**
>
> **A:** It is our oversight for being unfair. We were just following the common practice of latent stable diffusion, i.e., leveraging the pre-trained stable diffusion to ensure the stability of training and the diversity of its application scenarios.
>
> That being said, to gain a fair comparison, during the rebuttal days, without loading the checkpoint as the initialization, we re-train our model on the same COCO training set as baselines. The results are shown as follows, and we find our conclusions still hold water. The superiority of our proposed Salad model is still significant, especially on the COCO-A2I dataset.
>
> Result on COCO:
> | Model | FID | IS | CLIP|
> | :-----: | :----: | :----: | :----: |
> |Frido | 11.24 | 26.84 |70.46|
> | Ours | 10.57 | 28.16 |71.37|
>
> Result on COCO-A2I:
> | Model | FID | IS | CLIP|
> | :-----: | :----: | :----: | :----: |
> |Frido | 40.36|18.36|68.53|
> | Ours | 33. 61|24.16|70.34|
>
> ---
> **Q3:  The writing needs improvement. There are multiple typos throughout the paper and some inaccurate sentences.**
>
> **A:** We appreciate it much that you went through the paper carefully. We will carefully correct them all.
>
> ---
> **Q4:  What is the SGH module's output and the SIS step's input? Fig. 2 seems like you are inputting the scene graph matrix into the UNet, while Sec. 3.2 and Fig. 3 show that you convert the graphs into prompts and feed them to the attention layers.**
>
> **A:** Sorry for causing the confusion, and we will provide a clearer description in revision.
> 1) The output of the SGH module is three node matrices, i.e., object nodes $s_t^o$, the attributes nodes $s_t^a$, and relation nodes $s_t^r$, as shown in the right bottom of Fig 2. The value of three matrices is the label of each node. We can convert the three matrics into a scene graph without the need for any further calculations.
> 2) The SIS step takes as input a sequence of tokens derived from the scene graph at different semantic levels. For more details, please refer to Appendix B.2.
>
> ---
> **Q5:  Did you fine-tune the Stable Diffusion model by saying "For the SIS module, we load the parameters of Stable Diffusion3 (v1.4) as the initialization."?**
>
> **A:** Yes, we load the SD (version 1.4) weights, and then fine-tune it with our SGH module on the COCO data.
> The merits of employing the off-the-shelf checkpoint as the initialization are multifaceted. Firstly, based on the existing pre-trained parameters, we can achieve more stable training and faster convergence. Second, reusing the existing SD weights has been the default practice in the community, which also helps avoid repeating training the backbone from scratch.
>
> ---
> **Q6:  Why do all models achieve such high CLIP scores? Is the CLIP score the cosine similarity between image-text pairs or the CLIP R-precision metric?**
>
> **A:** The CLIP score [1] is defined as the cosine similarity between image-text pairs, as employed in the baselines. Specifically, for a generated image with visual CLIP embedding $v$ and an input prompt with textual CLIP embedding $c$, the CLIP score is computed as: $CLIP(v, c) = w * max(cos(v, c), 0)$, where $w$ is a re-scaling parameter. We used the officially-released code for accurate evaluation.
>
> [1] CLIPScore: A Reference-free Evaluation Metric for Image Captioning. EMNLP 2021

---

> ### Author Response · Authors · 2023-08-17
> **looking forward to your feedback**
>
> Dear Reviewer#DqEY,
>
> We would like to express our sincere appreciation for your efforts and valuable feedback.
> Your comments are essential to help us improve the quality of our work.
> To address your main concerns of COCO-A2I data construction and the experimental setup, we during rebuttal have run the experiments under a more reasonable setting, and presented the updated results.
> We kindly hope that you can take some time to check our response and re-evaluate our paper based on our replies.
> If you have any further concerns or questions, please do not hesitate to let us know. We will be happy to address them promptly.
>
> Best Regards.

---

> > ### Comment · Reviewer_DqEY · 2023-08-21
> > **Thank you for your response**
> >
> > Thank you for addressing some of my concerns. While I still have concerns about the experiment setup, I am on the fence and increased my rating to borderline accept.

---

> > > ### Author Response · Authors · 2023-08-21
> > >
> > > Dear Reviewer#DqEY,
> > >
> > > Thank you so much for your kind recognition of this work. All your feedback will be incorporated into revision.
> > > And if you still have concerns, please let us know. We would be more than willing to offer additional clarification.
> > >
> > > Best.

---

### Official Review · Reviewer_KZmg · 2023-07-11

**Soundness:** 4 excellent
**Presentation:** 4 excellent
**Contribution:** 3 good
**Rating:** 8
**Confidence:** 4

**Summary:**

This paper aims at the abstractive setting for text-to-image generation (T2I). They propose scene-graph hallucination (SGH) to perform imagination over the scene graph of the input prompt and make up the missing information. Then SGH can perform better T2I with the completed scene graph. Experiments on the COCO dataset demonstrate that T2I can significantly bridge the gap of abstract-to-intricate T2I.

**Strengths:**

+ This paper is well-written and easy to follow.
+ The goal of abstract-to-intricate T2I is important since we can not expect users always input their full prompt. This setting has the potential to improve practical applications.
+ The usage of scene graphs is effective and can lead to better imagination for T2I than the Chain-ot-Though (CoT) language prompt.
+ They provide detailed ablation studies from different aspects (Table 3/4 and Fig. 6/7) as well as rich qualitative examples (Fig. 5 and Fig. 12/13 in Appendix).

**Weaknesses:**

- Maybe I missed it, but what is the motivation for using scene graphs instead of LLM-completed prompts? Does the imagination over scene graphs work more robustly than LLM?
- Since they rely on imagination to deal with abstractive issues, the hallucination situation also raises. It will be better to evaluate this issue, and how is the trade-off between this and the T2I performance. How to avoid or mitigate this deficiency in the proposed framework?

**Questions:**

I do not have additional questions. Please see the Weakness and refer to other reviewers.

**Limitations:**

The hallucination issues over scene graphs should be carefully addressed.

---

> ### Author Rebuttal · Authors · 2023-08-09
>
> We would like to thank you for taking the time to provide valuable feedback! especially for the recognition of our work, such as 'well-written and easy to follow', ‘goal of abstract-to-intricate T2I is important’, and 'effective', as your support means a lot to us. Following, present the point-to-point response as follows to address your concerns.
>
> ---
>
> **Q1: Maybe I missed it, but what is the motivation for using scene graphs instead of LLM-completed prompts? Does the imagination over scene graphs work more robustly than LLM?**
>
>  **A:** The adoption of scene graphs over LLM-completed prompts is driven by two primary motivations. Firstly, employing scene graphs facilitates greater content controllability. LLM-completed prompts often introduce additional adjectives, attributives, or concatenate raw sentences to provide tangible explanations and contexts. However, such supplementary content inserted by LLMs lacks controllability, potentially leading to the inclusion of more abstract words, as exemplified by 'enthusiastic' and 'confident' in Figure 1, finally posing more challenges for downstream generative models. In contrast, scene graph-based imagination allows for enhanced visual controlling and a stable process, as the hallucination process entails expanding the initial scene graph with specific scene structures, where imaged objects, attributes, and relationships are drawn from limited pre-defined sets.
>
> Secondly, employing scene graphs reduces the difficulty of downstream image generation. Scene graphs offer a concise and precise means of depicting objects and their interrelationships, empowering fine-grained control over the semantic scene during the image generation process.
>
> ---
> **Q2: Since they rely on imagination to deal with abstractive issues, the hallucination situation also raises. It will be better to evaluate this issue, and how is the trade-off between this and the T2I performance. How to avoid or mitigate this deficiency in the proposed framework?**
>
>  **A:** As addressed in **Q1**, our scene graph imagination process aims to enrich the initial scene graphs by incorporating more specific scene structures, where objects, attributes, and relationships are selected from limited pre-defined sets, facilitating controlled imagination and mitigating undesired hallucination occurrences.
>
> To evaluate the proposed scene graph imagination module whether induce reasonable scene graphs (SGs), we actually have assessed the structure alignment, specifically the recall rate (TriRec.) of the ‘object-object’ pairs, between the induced SGs and the SGs of gold images known to represent reasonable scenes. Table 4 shows that 82.01% of 'object-object' pairs in the induced SGs exhibit high alignment with those in the gold SGs, signifying the capacity of our model to synthesize valid and coherent scene content.
>
> Moreover, the impact of the hallucination issue in T2I generation is relatively less significant compared to LLMs. In our T2I generation, hallucinations are relatively acceptable, particularly when dealing with abstractive scenarios, as demonstrated by experimental results. In contrast, LLMs, having no constraint, tends to produce more uncontrollable and undesirable content hallucinations. That being said, the hallucination issue does need to be addressed in some scenarios in our system, such as in the generation of images that emphasize real scenes, which is also a potential research direction for our further work.

---

### Author Rebuttal · Authors · 2023-08-09

# General Response to All Reviewers

Dear reviewers,

Thanks for all of your time to write valuable and constructive comments. Your feedback will definitely assist us in enhancing the quality of our paper, and thus we are committed to incorporating your suggestion in our revision process. Meanwhile, we feel encouraged so much that the reviewers find the new task setting is **'practical and important', 'vital' and 'interesting and meaningful'** (by reviewer #DqEY, #mzvT and #GuSJ), **the proposed method novel, interesting and effective** (by reviewer #QdRJ, #mzvT and #KZmg), and **our experiments solid and comprehensive** (by reviewer #QdRJ, #mzvT, #GuSJ and #DqEY). Your support means a lot to us!
At this juncture, we would like to re-emphasize the significance of this work.

While existing state-of-the-art (SoTA ) text-to-image (T2I) generation systems (e.g., Stable Diffusion) have shown incredible capability in creating high-quality images, the research on abstract-to-intricate (A2I) T2I has been largely overlooked, which can be also a very important setting in realist world. With such background, this work contributes to the following key aspects:

- We are the first to study the novel T2I setup of **intricate image synthesis from succinct abstract texts**, i.e., A2I T2I, for which we collect a dataset, COCO-A2I.
- We solve the A2I T2I with a novel scene graph (SG) hallucination framework implemented based on the discrete diffusion technique.
- We construct a diffusion-based T2I system, which employs the SG representations for highly controllable and scalable image generation.
- Our system empirically shows great advantages over existing SoTA baselines in the A2I T2I generation.

As recognized by reviewer#mzvT, the paper _'studies the specific conditioned image synthesis from an interesting and realistic perspective, with robust and novel technical methods'_, and _'will have the potential to unlock interesting future research'_. We much believe this work will show a broad impact on the future research of the community. Thus, we will release all the codes and resources upon acceptance.

In response to the reviewers' comments, we have thoroughly reviewed our paper, performed additional experiments, and prepared a comprehensive response. We will fix all the typos and improve the manuscript according to your comments. We hope that our paper adequately addresses your concerns. We also kindly hope reviewer #DqEY and reviewer  #GuSJ (both with borderline rejection) can raise the evaluation if our responses can effectively address the concerns, and look forward to your recognition.

Best regards.

---

### Author Response · Authors · 2023-08-15
**Sincerely looking forward to your feedback**

Dear reviewers,

Thanks again for all your prior efforts in reviewing this paper, and providing valuable feedback. We have carefully addressed all of your concerns and questions in our responses.
We kindly hope that you take some time to check our replies, and re-evaluate the paper based on our responses.
If you have any additional concerns or questions, please do not hesitate to let us know. We are committed to promptly addressing any further inquiries.

Best regards.

---

### Decision · Program_Chairs · 2023-09-21

**Decision:**

Accept (poster)

**Comment:**

The AC has carefully read the paper, reviews, author response, and the discussions. This paper aims at the abstractive setting for text-to-image generation. They propose scene-graph hallucination (SGH) to perform imagination over the scene graph of the input prompt and make up the missing information. All reviewers are positive towards acceptance. The AC agrees with the reviewers that this is a solid submission for NeurIPS and thus recommends acceptance with a certain confidence.